# EFFICIENT CONFORMAL PREDICTION WITH ORDER-PRESERVING PREDICTIONS FOR CLASSIFIERS

## ABSTRACT

Conformal prediction provides prediction sets with distribution-free, finite-sample coverage guarantees for machine learning classifiers. Numerous methods reduce set size by retraining classifiers or designing novel non-conformity scores, but they often suffer from high computational cost or inflexibility. To address this issue, we propose **Flexible Prediction Sets (FPS)**, a post-hoc framework that learns an order-preserving transformation which preserves the order of model's predicted class-probability while reshaping their magnitudes, enabling smaller conformal prediction sets. This transformation is obtained by optimizing a smooth surrogate of the set-size objective on a tuning dataset, then applied to the predicted class-probability before conformal calibration. This process yields smaller prediction sets while maintaining the coverage level. Theoretically, we prove coverage preservation under transformation, provide generalization bounds for the function class and surrogate risk, and show convergence to a stationary point. Empirically, extensive experiments on image and text benchmarks with multiple base machine learning classifiers demonstrate consistent reductions in set size at various nominal coverage rates, outperforming conformal prediction baselines.

## 1 INTRODUCTION

Uncertainty quantification is essential for reliable machine learning. In high-stakes settings such as medical diagnosis (Lambert et al., 2024), autonomous driving (Kendall & Gal, 2017), and risk-sensitive decision making in finance (Blasco et al., 2024), small predictive errors can lead to large costs or safety hazards. A broad toolkit has emerged for quantifying uncertainty, including confidence calibration (Guo et al., 2017), MC-Dropout (Gal & Ghahramani, 2016), deep ensembles (Lakshminarayanan et al., 2017) and conformal prediction (Vovk et al., 2005; Shafer & Vovk, 2008; Balasubramanian et al., 2014). Among these approaches, conformal prediction (CP) stands out for offering distribution-free, finite-sample coverage guarantees. In the classification setting (Sadinle et al., 2019; Romano et al., 2020; Angelopoulos et al., 2021), CP assembles a label set for each input with marginal coverage at the user-specified level.

A key goal in conformal prediction for classification is set-size efficiency: prediction sets that are small yet still achieve the desired coverage convey more actionable information. Split conformal prediction (Papadopoulos et al., 2002; Vovk et al., 2005) computes non-conformity scores on calibration data and selects a quantile threshold. At test time, it includes all labels below this threshold to ensure marginal coverage. Adaptive Prediction Sets (APS) (Romano et al., 2020) is a representative split CP method that defines the non-conformity score as the cumulative sum of probabilities needed to include the true label, with labels sorted by model-predicted probabilities. To further improve size efficiency, Regularized Adaptive Prediction Sets (RAPS) (Angelopoulos et al., 2021) introduces a refined non-conformity score with additional regularization, which stabilizes the threshold under heavy-tailed distributions. RAPS yields smaller sets with valid coverage but keeps predicted probabilities fixed, limiting flexibility and leaving potential gains untapped. This motivates directly changing the model-predicted class probabilities to improve set-size efficiency.

In this paper, we introduce Flexible Prediction Sets (FPS), a post-hoc framework designed to obtain smaller prediction sets in conformal prediction while maintaining the target coverage. The core of our approach is to apply an order-preserving transformation to the model's predicted class probabilities before the conformal prediction procedure. We specifically emphasize order preservation to ensure

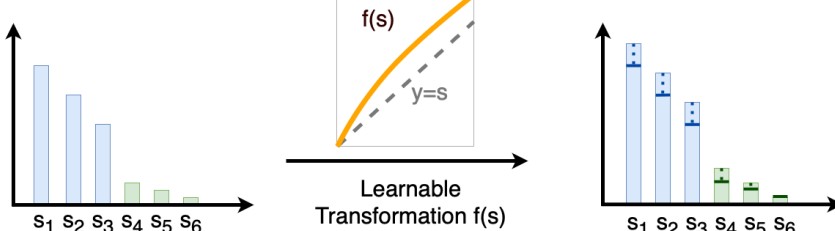

Figure 1: **Flexible Prediction Sets (FPS)**. The FPS framework introduces a learnable, order-preserving transformation $f(s)$ that rescales the predicted class probabilities before computing non-conformity scores. We illustrate one possibility of $f(s)$ which enlarges the separation between large probabilities (blue) and small ones (green). This transformation potentially makes the true label stand out, yielding smaller conformal prediction sets.

the base classifier's point prediction remains unchanged, thereby adhering to the post-hoc principle. Building upon this rule, our approach demonstrates that directly modifying predicted probabilities while preserving their original order is a powerful and flexible means of improving the size-efficiency of conformal prediction. As illustrated in Fig. 1, adjusting probability values while preserving their order enables FPS to produce more compact prediction sets.

Operationally, we learn the transformation that improves set size efficiency in two steps. First, we approximate it with a smooth, order-preserving parametric family whose derivative is positive by construction, implemented via an exponentiated trigonometric polynomial; second, we replace the hard indicators in the set-size objective with a sigmoid surrogate to enable stable, gradient-based optimization. After learning on an independent tuning set, we integrate the transformation into split conformal prediction. The only modification to the standard procedure is that the same order-preserving transformation is consistently applied to predicted class probabilities. It is first applied to the calibration set to compute non-conformity scores and determine the threshold, and then applied to each new input at test time to transform its predicted probabilities before forming the prediction set.

Our contributions are summarized as follows:

- **Order-preserving transform with learnable size objective.** We introduce a post-hoc framework that learns an order-preserving transformation of predicted class probabilities by minimizing a smooth surrogate within a parameterized function class, as implemented in Algorithm 2. When applied before split conformal prediction, the learned transform reduces prediction set size while maintaining valid coverage.

- **Extensive empirical validation.** As shown in Section 5, across diverse image and text classification benchmarks, our method consistently meets the target coverage while producing smaller prediction sets, thereby improving set-size efficiency. The advantage is empirically robust to the choice of backbone classifier and persists under dataset shifts, outperforming widely used conformal prediction baselines.

- **Theory for coverage, generalization, and optimization.** We establish that the learned transformation preserves the conformal prediction coverage guarantee, as shown in Theorem 1. Besides, we derive generalization bounds on the learned transformation obtained by minimizing the surrogate objective within the class of order-preserving functions, as presented in Theorem 2. Finally, we show in Theorem 3 that the optimization procedure converges to a stationary point in the sense of limit points.

## 2 RELATED WORK

**Conformal prediction.** Conformal prediction (CP) is a convenient uncertainty quantification framework that offers rigorous, distribution-free, finite-sample coverage guarantees (Vovk et al., 2005; Shafer & Vovk, 2008; Balasubramanian et al., 2014). It has been widely applied in regression (Papadopoulos et al., 2002; Lei et al., 2018; Romano et al., 2019) and classification (Sadinle et al., 2019; Romano et al., 2020; Bates et al., 2021), as well as in domain applications including medical imaging (Lu et al., 2022a;b), computer vision (Timans et al., 2024; Angelopoulos et al., 2021),

robotics control (Dixit et al., 2023; Sun et al., 2023), and natural language processing (Maltoudoglou et al., 2020; Choubey et al., 2022; Kumar et al., 2023; Quach et al., 2024). We study classification under split CP framework (Romano et al., 2020; Angelopoulos & Bates, 2021) with the goal of minimizing prediction set size while preserving the desired coverage level.

**Size efficiency.** While CP guarantees coverage, an equally important key performance criterion is size efficiency: the ability to produce small, and informative prediction sets (Sadinle et al., 2019; Romano et al., 2019; Dhillon et al., 2024; Gasparin & Ramdas, 2025). There is a substantial body of work on reducing the size of conformal classification sets. Existing approaches can be broadly grouped into two strands: (i) retraining methods that add size regularizers to the learning objective and train (or fine-tune) the classifier to reduce the prediction-set size (Yang & Kuchibhotla, 2021; Fisch et al., 2021; Einbinder et al., 2022; Stutz et al., 2022; Bai et al., 2022; Liang et al., 2023; Kiyani et al., 2024; Shi et al., 2025); and (ii) post-hoc procedures that keep the base predictor fixed and adjust the non-conformity scores to obtain tighter sets (Romano et al., 2020; Angelopoulos et al., 2021; Ghosh et al., 2023; Huang et al., 2024; Xi et al., 2024; Luo & Zhou, 2025). Our approach further explores the post-hoc line, avoiding the computational burden of retraining. Unlike fixed-form methods, we tune a flexible order-preserving transform that directly minimizes expected set size.

## 3 PRELIMINARIES

This section formalizes the multiclass classification problem and introduces the split conformal prediction framework for constructing prediction sets.

**K-class classification.** Let $(X, Y)$ be a random pair with $X \in \mathcal{X} \subset \mathbb{R}^d$ and $Y \in \mathcal{Y} = \{1, \ldots, K\}$. Assume we are given a black-box classifier that outputs predicted class-probability $\hat{\pi}_y(x)$ approximating $\mathbb{P}(Y = y \mid X = x)$ for each $y \in \mathcal{Y}$. The prediction rule for classification problems is $\hat{y} = \arg\max_{y \in \mathcal{Y}} \hat{\pi}_y(x)$. Throughout the paper, we assume that the predicted class-probability $\hat{\boldsymbol{\pi}}(x) = (\hat{\pi}_1(x), \ldots, \hat{\pi}_K(x))$ is standardized: for all $x$ and $y$, $0 \leq \hat{\pi}_y(x) \leq 1$ and $\sum_{y=1}^{K} \hat{\pi}_y(x) = 1$.

**Split conformal prediction.** For a user defined miscoverage rate $\alpha \in (0, 1)$, conformal prediction framework constructs a set-valued predictor $\mathcal{C} : \mathcal{X} \to 2^{\mathcal{Y}}$ that outputs a label set $\mathcal{C}(x) \subseteq \{1, \ldots, K\}$ with marginal coverage $\mathbb{P}\{Y \in \mathcal{C}(X)\} \geq 1 - \alpha$.

Split conformal prediction uses a calibration set $\mathcal{D}_{\text{cal}} = \{(x_i, y_i)\}_{i=1}^{n}$, where $(x_i, y_i) \overset{\text{i.i.d.}}{\sim} P_{XY}$ and $\mathcal{D}_{\text{cal}}$ is independent of the data used to fit the base classifier, together with a non-conformity score $E : \mathcal{X} \times \mathcal{Y} \to \mathbb{R}$. In CP procedure, compute scores $e_i = E(x_i, y_i)$ for $i = 1, \ldots, n$, and set the empirical $(1 - \alpha)$ quantile as the threshold:

$$\tau = \inf \left\{ e : \frac{\left| \{ i : E(x_i, y_i) \leq e \} \right|}{n} \geq \frac{\lceil (n+1)(1-\alpha) \rceil}{n} \right\}. \tag{1}$$

For a new input $x_{n+1}$, the prediction set is $\mathcal{C}(x_{n+1}, \tau) = \{ y \in \mathcal{Y} : E(x_{n+1}, y) \leq \tau \}$. We focus on the size efficiency of split-conformal methods that use an accumulated-output non-conformity score function. Two popular methods in this family are Adaptive Prediction Sets (APS) and Regularized Adaptive Prediction Sets (RAPS) (Romano et al., 2020; Angelopoulos et al., 2021).

In APS, the non-conformity score is the cumulative sum of sorted class-probability prediction up to the order of $y$:

$$E_{\text{APS}}(x, y, u) = \hat{\pi}_{(1)}(x) + \hat{\pi}_{(2)}(x) + \cdots + u \cdot \hat{\pi}_{(o(y,x))}(x).$$

RAPS augments APS with an order penalty to discourage inclusion of low–ordered labels:

$$E_{\text{RAPS}}(x, y, u) = E_{\text{APS}}(x, y, u) + \lambda \cdot \left( o(y, x) - k_{\text{reg}} \right)^+.$$

Here, $(\hat{\pi}_{(1)}(x) \geq \cdots \geq \hat{\pi}_{(K)}(x))$ are the model's predicted class-probability sorted in descending order; $o(y, x)$ is the order of $\hat{\pi}_y(x)$; $\lambda$ is the penalty applied to labels with order exceeding $k_{\text{reg}}$; and $u \sim \text{Unif}[0, 1]$ is a randomizer used at calibration to ensure exact finite-sample $(1 - \alpha)$ coverage, see Romano et al. (2020); Angelopoulos et al. (2021) for details.

---

**Algorithm 1** Find Threshold $\tau$ and Corresponding Calibration Components

---

**Input:** Calibration set $\mathcal{D}^c = \{(x_i, y_i)\}_{i=1}^n$; target $\alpha \in (0, 1)$; order-preserving function $f$.
**Precompute:** For each $(x_i, y_i) \in \mathcal{D}^c$, form the predicted class-probability vector $(s_{i,1}, \ldots, s_{i,K})$.
**Procedure:**
  1: **for** $i \in \{1, \ldots, n\}$ **do**
  2:     $r_i \leftarrow o(y_i, x_i)$;                $\triangleright$ order of the true label in the same order used in $(s_{i,1}, \ldots, s_{i,K})$
  3:     $e_i \leftarrow \displaystyle\sum_{j=1}^{r_i} f(s_{i,j})$;
  4: **end for**
  5: $k^\star \leftarrow \lceil (n+1)(1-\alpha) \rceil$;   $\tau \leftarrow e_{(k^\star)}$;
  6: Find $i^\star$ with $e_{i^\star} = \tau$;   $q^\star \leftarrow r_{i^\star}$;                    $\triangleright$ break ties randomly
  7: **Return** $\tau$ and $\boldsymbol{s}^c \leftarrow (s_{i^\star, 1}, \ldots, s_{i^\star, q^\star})$.
**Output:** Threshold $\tau$ and a $q^\star$-dimensional vector $\boldsymbol{s}^c$.

---

# 4   FLEXIBLE PREDICTION SETS

This section shows how to integrate the proposed FPS framework into the CP procedure and then develops a data-driven strategy to select the order-preserving function. Let $\boldsymbol{s}(x) = (s_1(x), \ldots, s_K(x))$ be the vector of predicted class-probability fed into the CP procedure. As an example, under the APS method we take $\boldsymbol{s}(x) = (\hat{\pi}_{(1)}(x), \ldots, \hat{\pi}_{(K)}(x))$, where $\hat{\pi}_{(1)}(x) \geq \cdots \geq \hat{\pi}_{(K)}(x)$. We denote by $f$ the transformation function, which acts componentwise on $\boldsymbol{s}$. Then, the non-conformity score after our transformation becomes:

$$E_{\text{FPS}}(x, y, u) = f(s_1(x)) + \cdots + u \cdot f(s_{o(y,x)}(x)), \tag{2}$$

where $o(y, x)$ is the index of the entry of $\boldsymbol{s}(x)$ that corresponds to class $y$.

**Set-size objective.** The conformal set size for a test input point $x$ can be written as

$$\text{len}(\mathcal{C}(x)) = \max\left\{ k : \sum_{i=1}^k f(s_i) \leq \tau \right\} = \sum_{k=1}^K \mathbb{1}\left\{ \sum_{i=1}^k f(s_i) \leq \sum_{i=1}^q f(s_i^c) \right\}.$$

Here, $\tau = \sum_{i=1}^q f(s_i^c)$ is the sample $(1 - \alpha)$-quantile of the calibration non-conformity scores, realized by some calibration example as in Algorithm 1.

**Order-preserving function.** Another important topic is to specify the function class for $f$. We follow a post-hoc principle: the base classifier's weights are fixed, and only its predicted class probabilities are reshaped. If $f$ is monotone and applied identically to all coordinates, the order of predicted class probabilities is preserved. Consequently, the $\arg\max$ label, and therefore the base classifier's point prediction, remains unchanged. This respects the model's property that *higher predicted probability means stronger preference*. Since $s$ is derived from probabilities, we consider the nonnegative, bounded regime $s \in [0, D]$ and set $f(0) = 0$ to anchor the scale. Formally, we restrict attention to the continuous differentiable and monotone function class:

$$\mathcal{F} = \big\{ f \in C^1([0, D]) : \forall\, s_1 < s_2 \Rightarrow f(s_1) < f(s_2),\, f(0) = 0 \big\}.$$

**Learning methods.** We make the set-size objective learnable via two mechanisms.

First, we approximate $\mathcal{F}$ by restricting to a parameterized subspace $\mathcal{G} \subseteq \mathcal{F}$:

$$\mathcal{G} = \left\{ g_{\mathbf{a}} \in C^1([0, D]) \,\middle|\, \begin{array}{l} \dfrac{\partial g_{\mathbf{a}}(s)}{\partial s} = \exp\Big(a_0 + \displaystyle\sum_{m=1}^M \big(a_{2m-1}\sin(ms) + a_{2m}\cos(ms)\big)\Big), \\[2mm] g_{\mathbf{a}}(0) = 0 \end{array} \right\},$$

with $\mathbf{a} = (a_0, \ldots, a_{2M}) \in \mathbb{R}^{2M+1}$. We adopt trigonometric polynomials for three significant reasons. (i) Structural guarantee: the exponential parameterization of the derivative $\partial g_{\mathbf{a}}(s)/\partial s$ enforces positivity and therefore preserves order. (ii) Approximation power: according to the Stone-Weierstrass

theorem (Stone, 1948; Rudin, 1987), the class $\mathcal{G}$ can uniformly approximate any continuous function on $[0, D]$. (iii) Optimization stability: the trigonometric polynomial parameterization is simple and its gradients are easy to compute, which yields a more direct and stable optimization procedure than approaches that enforce monotonicity by constraining parameters or outputs, including neural network parameterizations with positivity constraints. Monotone splines (Ramsay, 1988; He & Shi, 1998) are another option, but they require choosing both the number and the locations of knots and maintaining global monotonicity, which introduces inequality constraints or reparameterizations and complicates training. Performance is sensitive to knot placement and boundary treatment. Overall, our choice balances theoretical guarantees, flexibility, and practical stability.

The main hyperparameter introduced by class $\mathcal{G}$ is the degree $M$ of the trigonometric polynomial. A larger $M$ reduces approximation bias and captures finer structure, but it also increases the risk of overfitting. In practice, a moderate degree, for example $M \leq 5$, already performs well on image benchmarks. See Sec. 5.3 and Appendix A for sensitivity analysis and further details.

Second, to enable gradient-based learning, we replace the indicators with a smooth surrogate (e.g., a sigmoid function; (Stutz et al., 2022)) and optimize the resulting objective. Finally, the optimization problem reduces to the following form

$$\min_{g_{\mathbf{a}} \in \mathcal{G}} \mathcal{L}(g_{\mathbf{a}}), \ \mathcal{L}(g_{\mathbf{a}}) := \mathbb{E}\left[ \sum_{k=1}^{K} \sigma\Big( \beta^{-1} \big\{ \sum_{i=1}^{q} g_{\mathbf{a}}(s_i^c) - \sum_{i=1}^{k} g_{\mathbf{a}}(s_i) \big\} \Big) \right], \tag{3}$$

where $\sigma(x) = \frac{1}{1+e^{-x}}$ is the sigmoid function and $\beta > 0$ is a temperature parameter. The temperature rescales the margin inside $\sigma(\beta^{-1}\{\cdot\})$, thereby controlling the surrogate's smoothness. Smaller $\beta$ yields a sharper, indicator-like surrogate that preserves boundaries but risks tail gradient saturation and instability. Larger $\beta$ smooths transitions and eases optimization but loosens the approximation. In our experiments we fix $\beta = 1$, which we find strikes a practical balance between fidelity and trainability. Moreover, performance is relatively insensitive to changes in $\beta$ compared with other hyperparameters in most cases (see Sec. 5.3 for details).

Algorithm 2 describes the procedure for estimating the coefficients $\mathbf{a}$ using a parameter tuning set $\mathcal{D}_{\text{tune}}$ drawn i.i.d. from the same distribution as the black-box model's training set and the conformal calibration set. When learning $f$, we compute $\tau$ and the associated vector $\mathbf{s}^c$ with the randomizer $u = 1$ to make the training objective stable. At evaluation, we retain the standard split-conformal randomization to guarantee exact finite-sample $(1 - \alpha)$ coverage.

**Remark.** *The same idea extends to RAPS: it can be viewed as APS applied to a penalty–shifted predicted class-probability vector*

$$\boldsymbol{s}(x) := \big( \hat{\pi}_{(1)}(x), \ldots, \hat{\pi}_{(k_{\text{reg}})}(x), \ \hat{\pi}_{(k_{\text{reg}}+1)}(x) + \lambda, \ldots, \hat{\pi}_{(K)}(x) + \lambda \big),$$

*where $\hat{\pi}_{(1)}(x) \geq \cdots \geq \hat{\pi}_{(K)}(x)$ are the sorted model predictions and all orders exceeding $k_{\text{reg}}$ receive a constant shift $0 \leq \lambda \leq \Lambda$. Choosing $D \geq 1 + \Lambda$ keeps the shifted class probabilities remain in $[0, D]$, so $f \in C^1([0, D])$ is preserved. We then apply $f$ componentwise to transform $\boldsymbol{s}(x)$.*

# 5 EXPERIMENTS

In this section, we describe the experimental setup in Sec. 5.1, present the CP classification results under our FPS framework in Sec. 5.2, and analyze the sensitivity of the parameters in Sec. 5.3.

## 5.1 EXPERIMENTAL SETUP

**Datasets.** We evaluate on diverse classification benchmarks spanning images and text. For image classification, we use ImageNet (Deng et al., 2009) and ImageNet-V2 (Recht et al., 2019); for text classification, we use Banking77 (Casanueva et al., 2020; Lhoest et al., 2021; Muennighoff et al., 2022; Enevoldsen et al., 2025), an open-source dataset composed of online banking queries annotated with their corresponding intents. For each dataset, we randomly partition it into three disjoint parts in a $2 : 1 : 2$ ratio: a tuning set, a conformal calibration set, and an evaluation set. Furthermore, We randomly split the tuning set 1:1, half for gradient-based optimization, half for threshold selection. We also evaluated alternative data splits, details and results are provided in Appendix D.

---

**Algorithm 2** Tuning FPS Transformation

---

**Input:** Tuning set $\mathcal{D}_{\text{tune}} = \{(x_i, y_i)\}_{i=1}^{n_{\text{tune}}}$; temperature $\beta$; learning rate $\gamma$; maximum iterations $T$; tolerance $\varepsilon$; initial coefficients $\mathbf{a}^0 = (a_0^0, \ldots, a_{2M}^0)$.

**Precompute:** Randomly split $\mathcal{D}_{\text{tune}}$ into two sets: $\mathcal{D}$ for optimization and $\mathcal{D}^c$ for calibration. For each $(x_i, y_i) \in \mathcal{D}$, form the predicted class-probability vector $(s_{i,1}, \ldots, s_{i,K})$. For each $(x_i^c, y_i^c) \in \mathcal{D}^c$, form $(s_{i,1}^c, \ldots, s_{i,K}^c)$ likewise.

**Procedure:**

1: **for** $t = 1$ to $T$ **do**
2:      Find calibration components $\boldsymbol{s}^{c,t-1} = (s_1^{c,t-1}, \ldots, s_{q_{t-1}}^{c,t-1})$ by Algorithm 1 based on $g_{\mathbf{a}^{t-1}}$;
3:      Compute empirical loss from Eq. (3), with $n = |\mathcal{D}|$:

$$\mathcal{L}_n(g_{\mathbf{a}^{t-1}}, \boldsymbol{s}^{c,t-1}) \leftarrow \frac{1}{n} \sum_{i=1}^{n} \sum_{k=1}^{K} \sigma\Big(\beta^{-1}\big\{ \sum_{j=1}^{q_{t-1}} g_{\mathbf{a}^{t-1}}(s_j^{c,t-1}) - \sum_{j=1}^{k} g_{\mathbf{a}^{t-1}}(s_{i,j}) \big\}\Big);$$

4:      Update coefficients by one gradient step w.r.t. $\mathbf{a}$:

$$\mathbf{a}^t \leftarrow \mathbf{a}^{t-1} - \gamma \nabla_{\mathbf{a}} \mathcal{L}_n(g_{\mathbf{a}^{t-1}}, \boldsymbol{s}^{c,t-1}).$$

5:      **if** $\|\mathbf{a}^t - \mathbf{a}^{t-1}\|_2 < \varepsilon$ **then**
6:          **return** $\mathbf{a}^t$.                                  ▷ converged
7:      **end if**
8: **end for**
9: **return** $\mathbf{a}^T$.                                   ▷ maximum iterations reached

**Output:** Estimated coefficients $\widehat{\mathbf{a}}$. Prediction sets can then be constructed by CP algorithm using transformed predicted class probabilities $g_{\widehat{\mathbf{a}}}(\boldsymbol{s}(x))$.

---

**Base models.** For ImageNet and ImageNet-V2, we use eight off-the-shelf ImageNet-pretrained deep classifiers from the TorchVision (Paszke et al., 2019): ResNet101/152 (He et al., 2016), ResNeXt101 (Xie et al., 2017), DenseNet-161 (Huang et al., 2017), VGG-16 (Simonyan & Zisserman, 2015), and ShuffleNet (Zhang et al., 2018). For the Banking, we use publicly available Transformer encoders from the Hugging Face Hub (Wolf et al., 2020): BERT (Devlin et al., 2019), RoBERTa (Liu et al., 2019), DistilBERT (Sanh et al., 2019), and DistilRoBERTa (a distilled variant of RoBERTa). Across all experiments, base classifiers' weights are kept fixed.

**Conformal prediction.** We evaluate two target miscoverage levels, $\alpha \in \{0.05, 0.10\}$. For a fair comparison, we evaluate APS against its FPS transformed variant, using identical base classifier outputs generated under the same random seed. For RAPS, we select $(k_{\text{reg}}, \lambda)$ by a grid search following Angelopoulos et al. (2021). When comparing RAPS with its FPS transformed counterpart, we reuse the same $(k_{\text{reg}}, \lambda)$ and the same seeded classifier outputs, ensuring that any observed differences arise from the learned transformation $g_{\widehat{\mathbf{a}}}$ rather than from hyperparameters or randomness.

**Evaluation metrics.** Let $\mathcal{D}_{\text{eval}} = \{(x_i, y_i)\}_{i=1}^{n_{\text{eval}}}$ be the evaluation set. We report two quantities at target level $\alpha$: Coverage, the fraction of evaluation examples whose true label lies in the prediction set; and Size, the mean cardinality of the set. We target coverage very close to the nominal $1 - \alpha$; size is compared at matched coverage levels, where a smaller average size indicates higher efficiency.

$$\text{Coverage} = \frac{1}{n_{\text{eval}}} \sum_{i=1}^{n_{\text{eval}}} \mathbb{1}\{y_i \in \mathcal{C}(x_i)\}, \qquad \text{Size} = \frac{1}{n_{\text{eval}}} \sum_{i=1}^{n_{\text{eval}}} |\mathcal{C}(x_i)|.$$

## 5.2 MAIN RESULTS

We evaluate three datasets (ImageNet, ImageNet-V2, and Banking77), multiple base classifiers, and conformal prediction methods (APS, RAPS, and their FPS-transformed variants) at user-defined target levels $\alpha$. To quantify variability, we repeat 10 independent trials on ImageNet and 100 trials on ImageNet-V2 and Banking77, reporting Coverage (mean) and Size (mean $\pm$ standard error) across runs. All experiments are executed on a machine with an Intel Xeon CPU (12 cores) and two NVIDIA GeForce GTX 1080 Ti GPUs.

Table 1: Coverage and Size results on ImageNet across $\alpha$ levels and base image classifiers. APS and RAPS are baselines; +ours denotes applying our FPS framework (APS+ours, RAPS+ours).

| Model | $\alpha$ | Coverage | | | | Size | | | |
|---|---|---|---|---|---|---|---|---|---|
| | | APS | APS+ours | RAPS | RAPS+ours | APS | APS+ours | RAPS | RAPS+ours |
| ResNeXt101 | 0.05 | 0.951 | 0.951 | 0.950 | 0.948 | $20.865 \pm 0.342$ | $\mathbf{10.939} \pm 0.217$ | $3.829 \pm 0.080$ | $\mathbf{3.640} \pm 0.043$ |
| | 0.10 | 0.901 | 0.900 | 0.900 | 0.898 | $7.171 \pm 0.109$ | $\mathbf{2.894} \pm 0.033$ | $2.020 \pm 0.011$ | $\mathbf{1.966} \pm 0.009$ |
| ResNet152 | 0.05 | 0.951 | 0.950 | 0.950 | 0.950 | $14.725 \pm 0.186$ | $\mathbf{8.298} \pm 0.112$ | $4.087 \pm 0.032$ | $\mathbf{4.032} \pm 0.042$ |
| | 0.10 | 0.900 | 0.901 | 0.901 | 0.900 | $6.360 \pm 0.065$ | $\mathbf{3.010} \pm 0.039$ | $2.260 \pm 0.006$ | $\mathbf{2.176} \pm 0.012$ |
| ResNet101 | 0.05 | 0.951 | 0.950 | 0.949 | 0.949 | $16.091 \pm 0.130$ | $\mathbf{9.022} \pm 0.175$ | $4.417 \pm 0.063$ | $\mathbf{4.382} \pm 0.052$ |
| | 0.10 | 0.902 | 0.901 | 0.900 | 0.898 | $7.015 \pm 0.057$ | $\mathbf{3.315} \pm 0.033$ | $2.387 \pm 0.013$ | $\mathbf{2.286} \pm 0.010$ |
| DenseNet161 | 0.05 | 0.950 | 0.951 | 0.949 | 0.949 | $17.218 \pm 0.184$ | $\mathbf{9.866} \pm 0.140$ | $4.702 \pm 0.104$ | $\mathbf{4.664} \pm 0.080$ |
| | 0.10 | 0.901 | 0.900 | 0.898 | 0.900 | $6.956 \pm 0.101$ | $\mathbf{3.275} \pm 0.039$ | $2.338 \pm 0.020$ | $\mathbf{2.299} \pm 0.011$ |
| VGG16 | 0.05 | 0.949 | 0.949 | 0.951 | 0.950 | $23.917 \pm 0.367$ | $\mathbf{15.329} \pm 0.194$ | $8.803 \pm 0.548$ | $\mathbf{8.542} \pm 0.380$ |
| | 0.10 | 0.899 | 0.899 | 0.899 | 0.898 | $11.845 \pm 0.086$ | $\mathbf{5.943} \pm 0.051$ | $3.768 \pm 0.012$ | $\mathbf{3.577} \pm 0.019$ |
| ShuffleNet | 0.05 | 0.949 | 0.950 | 0.950 | 0.950 | $54.133 \pm 1.072$ | $\mathbf{27.588} \pm 0.521$ | $15.696 \pm 0.719$ | $\mathbf{15.029} \pm 0.460$ |
| | 0.10 | 0.899 | 0.901 | 0.900 | 0.899 | $22.584 \pm 0.305$ | $\mathbf{8.931} \pm 0.141$ | $5.026 \pm 0.077$ | $\mathbf{4.898} \pm 0.068$ |

Table 2: Coverage and Size results on ImageNet-V2 across $\alpha$ levels and base image classifiers.

| Model | $\alpha$ | Coverage | | | | Size | | | |
|---|---|---|---|---|---|---|---|---|---|
| | | APS | APS+ours | RAPS | RAPS+ours | APS | APS+ours | RAPS | RAPS+ours |
| ResNeXt101 | 0.05 | 0.951 | 0.951 | 0.951 | 0.950 | $72.310 \pm 0.756$ | $\mathbf{50.075} \pm 0.402$ | $19.746 \pm 0.449$ | $\mathbf{18.656} \pm 0.323$ |
| | 0.10 | 0.901 | 0.900 | 0.900 | 0.900 | $27.597 \pm 0.283$ | $\mathbf{14.436} \pm 0.103$ | $6.163 \pm 0.120$ | $\mathbf{5.933} \pm 0.088$ |
| ResNet152 | 0.05 | 0.951 | 0.951 | 0.950 | 0.949 | $42.745 \pm 0.403$ | $\mathbf{35.740} \pm 0.292$ | $16.173 \pm 0.361$ | $\mathbf{15.277} \pm 0.246$ |
| | 0.10 | 0.900 | 0.901 | 0.900 | 0.900 | $17.869 \pm 0.160$ | $\mathbf{12.375} \pm 0.092$ | $5.794 \pm 0.081$ | $\mathbf{5.625} \pm 0.054$ |
| ResNet101 | 0.05 | 0.950 | 0.949 | 0.951 | 0.950 | $48.963 \pm 0.459$ | $\mathbf{39.948} \pm 0.311$ | $21.691 \pm 0.496$ | $\mathbf{20.029} \pm 0.391$ |
| | 0.10 | 0.899 | 0.899 | 0.901 | 0.900 | $20.937 \pm 0.175$ | $\mathbf{14.218} \pm 0.099$ | $6.957 \pm 0.119$ | $\mathbf{6.618} \pm 0.071$ |
| DenseNet161 | 0.05 | 0.950 | 0.949 | 0.951 | 0.950 | $54.296 \pm 0.641$ | $\mathbf{43.468} \pm 0.416$ | $22.168 \pm 0.503$ | $\mathbf{20.601} \pm 0.343$ |
| | 0.10 | 0.899 | 0.900 | 0.902 | 0.901 | $20.776 \pm 0.237$ | $\mathbf{13.437} \pm 0.118$ | $6.825 \pm 0.093$ | $\mathbf{6.575} \pm 0.071$ |
| VGG16 | 0.05 | 0.950 | 0.950 | 0.950 | 0.949 | $57.578 \pm 0.483$ | $\mathbf{51.266} \pm 0.415$ | $40.564 \pm 9.667$ | $\mathbf{29.368} \pm 0.507$ |
| | 0.10 | 0.900 | 0.900 | 0.898 | 0.899 | $27.740 \pm 0.198$ | $\mathbf{21.620} \pm 0.139$ | $11.824 \pm 0.190$ | $\mathbf{11.346} \pm 0.128$ |
| ShuffleNet | 0.05 | 0.950 | 0.951 | 0.950 | 0.949 | $130.688 \pm 1.112$ | $\mathbf{113.189} \pm 0.767$ | $74.546 \pm 1.359$ | $\mathbf{71.162} \pm 1.080$ |
| | 0.10 | 0.900 | 0.900 | 0.899 | 0.899 | $59.976 \pm 0.550$ | $\mathbf{39.292} \pm 0.293$ | $23.361 \pm 0.481$ | $\mathbf{22.463} \pm 0.360$ |

As shown in Tables 1, 2, and 3, our FPS transformation reduces set size for both APS and RAPS while maintaining coverage, consistently across $\alpha$ levels, base classifiers, and multi-modal datasets. Since our method is post-hoc and computationally light, we fix $\beta = 1$ and select the hyperparameters $M$ and $\gamma$ via a simple grid search. Implementation details are provided in Appendix A.

## 5.3 SENSITIVITY ANALYSIS

We conduct a sensitivity analysis for all the parameters introduced by FPS: the sigmoid temperature $\beta$, the trigonometric polynomial order $M$, and the learning rate $\gamma$. For each factor, we use a grid of values while holding the remaining hyperparameters fixed, tune the FPS transformation for both APS and RAPS, evaluate the resulting conformal prediction sets Size and Coverage at $\alpha \in \{0.05, 0.10\}$. As seen in Table 4, prediction-set size is relatively more sensitive to $\gamma$ and $M$ than to $\beta$, which corroborates our hyperparameter selection strategy of fixing $\beta = 1$ while tuning $M$ and $\gamma$. Table 5 indicates that the target nominal coverage is achieved irrespective of the hyperparameter configuration. Implementation details are also provided in Appendix A.

## 6 THEORETICAL RESULTS

This section provides the theoretical guarantees for our proposed FPS method. Theorem 1 shows that split CP procedure, after the FPS transformation, still preserves the coverage guarantee. Theorem 2 characterizes the generalization bound of the approximation approach used in FPS. Finally, Theorem 3

Table 3: Coverage and Size results on Banking77 across $\alpha$ levels and base text classifiers.

| Model | $\alpha$ | Coverage | | | | Size | | | |
|---|---|---|---|---|---|---|---|---|---|
| | | APS | APS+ours | RAPS | RAPS+ours | APS | APS+ours | RAPS | RAPS+ours |
| BERT | 0.05 | 0.950 | 0.949 | 0.948 | 0.948 | $2.537 \pm 0.031$ | $\mathbf{1.577 \pm 0.034}$ | $1.446 \pm 0.014$ | $\mathbf{1.361 \pm 0.015}$ |
| | 0.10 | 0.898 | 0.900 | 0.898 | 0.898 | $1.524 \pm 0.011$ | $\mathbf{0.972 \pm 0.001}$ | $1.153 \pm 0.009$ | $\mathbf{0.972 \pm 0.001}$ |
| RoBERTa | 0.05 | 0.949 | 0.950 | 0.950 | 0.949 | $2.017 \pm 0.030$ | $\mathbf{1.273 \pm 0.013}$ | $1.242 \pm 0.010$ | $\mathbf{1.186 \pm 0.013}$ |
| | 0.10 | 0.898 | 0.899 | 0.902 | 0.898 | $1.316 \pm 0.009$ | $\mathbf{0.967 \pm 0.001}$ | $1.082 \pm 0.007$ | $\mathbf{0.966 \pm 0.001}$ |
| DistilBERT | 0.05 | 0.949 | 0.950 | 0.950 | 0.950 | $2.127 \pm 0.023$ | $\mathbf{1.449 \pm 0.015}$ | $1.461 \pm 0.012$ | $\mathbf{1.344 \pm 0.010}$ |
| | 0.10 | 0.900 | 0.901 | 0.902 | 0.899 | $1.463 \pm 0.009$ | $\mathbf{0.977 \pm 0.001}$ | $1.191 \pm 0.007$ | $\mathbf{0.977 \pm 0.001}$ |
| DistilRoBERTa | 0.05 | 0.949 | 0.948 | 0.950 | 0.949 | $4.226 \pm 0.041$ | $\mathbf{2.080 \pm 0.020}$ | $2.299 \pm 0.023$ | $\mathbf{1.842 \pm 0.013}$ |
| | 0.10 | 0.898 | 0.898 | 0.900 | 0.899 | $2.729 \pm 0.028$ | $\mathbf{1.647 \pm 0.134}$ | $1.703 \pm 0.013$ | $\mathbf{1.123 \pm 0.020}$ |

Table 4: Size sensitivity for FPS at $\alpha \in \{0.05, 0.10\}$. Each hyperparameter is varied in turn, with the others held fixed as indicated; we report the mean size over repeated experiments. For each hyperparameter we also report the range $\Delta$ (max–min) over its four settings.

| $\alpha$ | Method | *Vary* $\beta$ ($\gamma$=0.001, $M$=1) | | | | | *Vary* $M$ ($\beta$=1, $\gamma$=0.001) | | | | | *Vary* $\gamma$ ($\beta$=1, $M$=1) | | | | |
|---|---|---|---|---|---|---|---|---|---|---|---|---|---|---|---|---|
| | | $\beta$=0.01 | $\beta$=0.1 | $\beta$=1 | $\beta$=10 | $\Delta_\beta$ | $M$=1 | $M$=2 | $M$=3 | $M$=4 | $\Delta_M$ | $\gamma$=$10^{-5}$ | $\gamma$=$10^{-4}$ | $\gamma$=$10^{-3}$ | $\gamma$=$10^{-2}$ | $\Delta_\gamma$ |
| 0.05 | APS+ours | 9.08 | 8.78 | 8.89 | 9.06 | **0.30** | 8.89 | 8.21 | 7.68 | 7.22 | **1.67** | 14.05 | 13.23 | 8.89 | 8.66 | **5.39** |
| | RAPS+ours | 4.11 | 4.04 | 4.05 | 4.16 | **0.12** | 4.05 | 4.09 | 6.19 | 6.83 | **2.78** | 4.12 | 4.04 | 4.05 | 4.03 | **0.09** |
| 0.10 | APS+ours | 4.99 | 4.81 | 4.75 | 4.78 | **0.24** | 4.75 | 3.39 | 3.03 | 3.09 | **1.72** | 6.26 | 6.14 | 4.75 | 3.05 | **3.21** |
| | RAPS+ours | 2.20 | 2.22 | 2.21 | 2.28 | **0.08** | 2.21 | 2.20 | 2.19 | 2.17 | **0.04** | 2.26 | 2.27 | 2.21 | 2.19 | **0.08** |

establishes that Algorithm 2 admits a subsequence converging to a stationary point. The complete proofs for all the theoretical results are given in Appendix E.

**Theorem 1** (FPS coverage guarantee). *Suppose $\{(x_i, y_i, u_i)\}_{i=1}^n$ and $(x_{n+1}, y_{n+1}, u_{n+1})$ are i.i.d. samples. Let $g_{\widehat{a}}$ be selected by Algorithm 2 using a tuning set $\mathcal{D}_{\text{tune}}$ of i.i.d. samples, which is independent of the conformal calibration and evaluation set. Let $\mathcal{C}_{g_{\widehat{a}}}(x, u, \tau)$ be the split CP prediction set obtained using the non-conformity score in Eq. (2) (with $f$ replaced by $g_{\widehat{a}}$) and the corresponding threshold $\tau$ defined in Eq. (1). Suppose further that $\mathcal{F}$ is a measurable function class. Then the following coverage guarantee holds:*

$$1 - \alpha \ \leq \ \mathbb{P}\big\{y_{n+1} \in \mathcal{C}_{g_{\widehat{a}}}(x_{n+1}, u_{n+1}, \tau)\big\} \ \leq \ 1 - \alpha + \frac{1}{n+1}.$$

Theorem 1 implies that FPS transformation preserves the coverage of the base CP method.

Prior to further analysis, we let $g_{\widehat{a}} \in \mathcal{G}$ denote the transformation returned by Algorithm 2, $f^\star \in \mathcal{F}$ be a minimizer of the population loss: $f^\star \in \arg\min_{f \in \mathcal{F}} \mathcal{L}(f)$, and define the empirical version of $\mathcal{L}(g_{\mathbf{a}})$ appearing in Eq. (3): $\mathcal{L}_n(g_{\mathbf{a}}) = \frac{1}{n} \sum_{i=1}^n \sum_{k=1}^K \sigma\Big(\beta^{-1}\big\{\sum_{j=1}^q g(s_j^c) - \sum_{j=1}^k g(s_{i,j})\big\}\Big)$.

**Assumption 1** (Approximate empirical risk minimization). *Let $g_{\widehat{a}}$ be the transformation function returned by Algorithm 2, assume*

$$\mathcal{L}_n(g_{\widehat{a}}) \ \leq \ \inf_{g_{\mathbf{a}} \in \mathcal{G}} \mathcal{L}_n(g_{\mathbf{a}}) \ + \ \varepsilon_{\text{opt}},$$

*where $\varepsilon_{\text{opt}} \geq 0$ is the optimization suboptimality for empirical risk.*

**Lemma 1** (Approximation error). *Define $\delta_M := \inf_{g \in \mathcal{G}} \|g - f^\star\|_\infty$. Then $\delta_M \to 0$ as $M \to \infty$.*

Equipped with Assumption 1 and Lemma 1, we show that the surrogate loss used by FPS, together with our function-space approximation scheme, admits a high-probability generalization bound. In particular, the excess risk $\mathcal{L}(g_{\widehat{a}}) - \mathcal{L}(f^\star)$ is controlled by a standard estimation term (scaling with $n$) plus an approximation term (scaling with $M$). We state the result Theorem 2 formally below.

**Theorem 2** (Generalization bound). *Assume $\|\mathbf{a}\|_1 \leq A$ and Assumption 1 holds. Then, for any $\delta \in (0, 1)$, with probability at least $1 - \delta$,*

$$\mathcal{L}(g_{\widehat{a}}) - \mathcal{L}(f^\star) \ \leq \ C_1 L_K \frac{e^A D}{\sqrt{n}} \ + \ C_2 \sqrt{\frac{\log(1/\delta)}{n}} \ + \ L_K \delta_M \ + \ \varepsilon_{\text{opt}},$$

Table 5: Coverage sensitivity for FPS at $\alpha \in \{0.05, 0.10\}$. Each hyperparameter is varied in turn, with the others held fixed as indicated; we report the mean coverage over repeated experiments.

| | | Vary $\beta$ ($\gamma$=0.001, $M$=1) | | | | Vary $M$ ($\beta$=1, $\gamma$=0.001) | | | | Vary $\gamma$ ($\beta$=1, $M$=1) | | | |
|---|---|---|---|---|---|---|---|---|---|---|---|---|---|
| $\alpha$ | Method | $\beta$=0.01 | $\beta$=0.1 | $\beta$=1 | $\beta$=10 | $M$=1 | $M$=2 | $M$=3 | $M$=4 | $\gamma$=$10^{-5}$ | $\gamma$=$10^{-4}$ | $\gamma$=$10^{-3}$ | $\gamma$=$10^{-2}$ |
| 0.05 | APS+ours | 0.949 | 0.950 | 0.950 | 0.951 | 0.950 | 0.949 | 0.949 | 0.949 | 0.949 | 0.949 | 0.950 | 0.951 |
| | RAPS+ours | 0.950 | 0.950 | 0.950 | 0.949 | 0.950 | 0.951 | 0.950 | 0.951 | 0.950 | 0.950 | 0.950 | 0.949 |
| 0.10 | APS+ours | 0.900 | 0.901 | 0.899 | 0.901 | 0.899 | 0.899 | 0.899 | 0.900 | 0.899 | 0.901 | 0.899 | 0.900 |
| | RAPS+ours | 0.899 | 0.902 | 0.901 | 0.898 | 0.901 | 0.902 | 0.902 | 0.899 | 0.899 | 0.902 | 0.901 | 0.899 |

where $L_K = \frac{K(3K+1)}{8\beta}$, $C_1, C_2 > 0$ are universal constants. Furthermore, if $n, M \to \infty$ with $\delta_M \to 0$, then by Lemma 1,

$$\mathcal{L}(g_{\hat{a}}) - \mathcal{L}(f^\star) = \varepsilon_{\text{opt}} + o_{\mathbb{P}}(1),$$

i.e., the excess risk is asymptotically controlled solely by the optimization error.

Finally, we explore the convergence of Algorithm 2 in practice. Updating the components $s^c$ on the calibration set may increase the loss, and we therefore state the following assumption.

**Assumption 2** (Vanishing loss update). *At iteration $t$ of Algorithm 2, we take a gradient step with calibration components frozen at $s^{c,t-1}$ to obtain $\mathbf{a}^t$, then refresh calibration components via $g_{\mathbf{a}^t}$ to get $s^{c,t}$. Assume there exists a nonnegative sequence $\{\delta_t\}_{t\geq1}$ with running average $\lim_{T\to\infty} \frac{1}{T}\sum_{t=1}^{T} \delta_t = 0$, such that for every $t$ the loss after calibration update satisfies*

$$\mathcal{L}_n(g_{\mathbf{a}^t}, s^{c,t}) \leq \mathcal{L}_n(g_{\mathbf{a}^t}, s^{c,t-1}) + \delta_t.$$

In our classification settings, Assumption 2 often holds. In particular, across iterations, the calibration components vector $s^{c,t}$ is not updated every time. Moreover, when it is updated, the change is small. This is because the vector is short on average and its large entries are concentrated in the first few coordinates, and supporting intuition and experimental evidence are provided in Appendix C. With Assumption 2 in place, we formally show that the sequence $\{\mathbf{a}^t\}_{t\geq1}$ generated by Algorithm 2 admits a stationary limit point.

**Theorem 3** (Limit point stationarity). *Assume Assumption 2 holds, $\|\mathbf{a}\|_1 \leq A$, and the fixed gradient step size satisfies $\gamma \in (0, 1/L_A]$ with $L_A = \frac{KDe^A}{24\beta}\left[\frac{De^A}{\beta}(14K^2 + 9K + 1) + 3(3K+1)\right]$. Then, for every $t \geq 1$, we have*

$$\mathcal{L}_n(g_{\mathbf{a}^t}, s^{c,t}) \leq \mathcal{L}_n(g_{\mathbf{a}^{t-1}}, s^{c,t-1}) - \frac{\gamma}{2}\left\|\nabla_{\mathbf{a}}\mathcal{L}_n(g_{\mathbf{a}^{t-1}}, s^{c,t-1})\right\|^2 + \delta_t.$$

*Since $\mathcal{L}_n$ is the empirical average of a finite sum sigmoid terms, we have $\mathcal{L}_n \geq 0$, consequently,*

$$\lim_{T\to\infty} \frac{1}{T}\sum_{t=1}^{T}\left\|\nabla_{\mathbf{a}}\mathcal{L}_n(g_{\mathbf{a}^t}, s^{c,t})\right\|^2 = 0, \qquad \liminf_{t\to\infty}\left\|\nabla_{\mathbf{a}}\mathcal{L}_n(g_{\mathbf{a}^t}, s^{c,t})\right\| = 0.$$

## 7 CONCLUSION AND FUTURE WORK

In this work, we introduced Flexible Prediction Sets (FPS), a post-hoc framework that improves the size efficiency of conformal prediction for classifiers. FPS applies an order-preserving transformation to predicted probabilities and, when integrated into standard conformal prediction procedures, yields smaller sets. We learn the transformation by optimizing a smooth surrogate of expected set size within an increasing function class. Across diverse image and text benchmarks, FPS reduces set sizes for APS and RAPS while maintaining target coverage, supported by proofs of coverage preservation, generalization bounds, and optimization convergence.

While our standard approach uses a held-out tuning set for theoretical rigor, it is data-intensive. Empirically, partially overlapping the tuning and calibration sets still yields valid coverage despite violating exchangeability, as shown in Appendix D. A promising direction for future work is to formally analyze FPS under data reuse. Another avenue for future work is to replace the length surrogate with alternative objectives that tailor FPS to different desiderata, for example targeting conditional coverage in applications where it is required.

ETHICS STATEMENT

This research is methodological, focusing on the development of a new framework, Flexible Prediction Sets (FPS), to improve the size efficiency of conformal prediction for machine learning classifiers. Our work does not involve human subjects, and therefore no Institutional Review Board (IRB) approval was required. All experiments were conducted on standard, publicly available benchmarks, which are widely used in the machine learning community. Our research does not involve the collection of new data, nor does it process personally identifiable or sensitive information, thus mitigating concerns related to data privacy and security.

REPRODUCIBILITY STATEMENT

To ensure the reproducibility of our work, we provide detailed descriptions of our theoretical results and experimental setup. The theoretical results presented in Section 6 are accompanied by complete mathematical proofs in Appendix E. Our full experimental setup is described in Section 5.1. Further implementation details, such as data splitting protocols and the specific hyperparameters used to obtain the results, are provided in Appendix A. The source code is provided in the supplementary material and will be made publicly available upon publication.

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

# Appendix

In the appendix, we present implementation details for the results in the main paper in Appendix A, and visualize some of the learned transformations in Appendix B. We also provide intuition and experimental evidence for Assumption 2, see Appendix C. In addition, we report results under a partially overlapping data-splits strategy, see Appendix D. Moreover, we provide detailed proofs for the theorems and lemmas in the main paper, see Appendix E. We further discuss FPS together with two related methods, Temperature Scaling and Least Ambiguous Sets, see Appendices F and G. Finally, we clarify the usage of large language models in our paper, see Appendix H.

## A IMPLEMENTATION DETAILS

In this section, we detail the experimental setup for the results reported in Tables 1-5.

### A.1 IMPLEMENTATION DETAILS OF TABLE 1

We split the 50k-sample ImageNet validation set into three disjoint subsets: 20k for tuning the FPS transformation (split evenly into 10k for gradient-based learning and 10k for searching the threshold and associated calibration components), 10k for conformal calibration, and 20k for evaluating the size and coverage of prediction sets. Following Angelopoulos et al. (2021), we select $k_{\text{reg}}$ and search $\lambda \in \{0.001, 0.005, 0.01, 0.02, 0.05\}$ for RAPS. With $\beta = 1$ fixed, we search $M \in \{1, 2, 3, 4\}$ and $\gamma \in \{10^{-2}, 10^{-3}, 10^{-4}, 10^{-5}\}$. We then tune FPS, initialized at zero $\mathbf{a}^0$, using AdamW (weight decay $10^{-4}$). The final hyperparameters are as follows. For $\alpha = 0.05$: APS+ours uses $M = 2$, $\gamma = 10^{-3}$; RAPS+ours uses $M = 1$, $\gamma = 10^{-4}$. For $\alpha = 0.10$: APS+ours uses $M = 3$, $\gamma = 10^{-3}$; RAPS+ours uses $M = 2$, $\gamma = 10^{-3}$. We repeat each experiment 10 times and report the mean size with its standard error, and the mean coverage.

### A.2 IMPLEMENTATION DETAILS OF TABLE 2

We split the 10k-sample ImageNetV2 set into three disjoint subsets: 4k for tuning the FPS transformation (split evenly into 2k for gradient-based learning and 2k for searching the threshold and associated calibration components), 2k for conformal calibration, and 4k for evaluating the size and coverage of prediction sets. Following Angelopoulos et al. (2021), we select $k_{\text{reg}}$ and search $\lambda \in \{0.001, 0.005, 0.01, 0.02, 0.05\}$ for RAPS. With $\beta = 1$ fixed, we search $M \in \{1, 2, 3, 4\}$ and $\gamma \in \{10^{-2}, 10^{-3}, 10^{-4}, 10^{-5}\}$. We then tune FPS, initialized at zero $\mathbf{a}^0$, using AdamW (weight decay $10^{-4}$). The final hyperparameters are as follows. For $\alpha = 0.05$: APS+ours uses $M = 1$, $\gamma = 10^{-3}$; RAPS+ours uses $M = 1$, $\gamma = 10^{-4}$. For $\alpha = 0.10$: APS+ours uses $M = 3$, $\gamma = 10^{-3}$; RAPS+ours uses $M = 1$, $\gamma = 10^{-3}$. We repeat each experiment 100 times and report the mean size with its standard error, and the mean coverage.

### A.3 IMPLEMENTATION DETAILS OF TABLE 3

We split the 3076-sample Banking77 test set into three disjoint subsets: 1230 for tuning the FPS transformation (split evenly into 615 for gradient-based learning and 615 for searching the threshold and associated calibration components), 615 for conformal calibration, and 1231 for evaluating the size and coverage of prediction sets. Following Angelopoulos et al. (2021), we select $k_{\text{reg}}$ and search $\lambda \in \{0.001, 0.005, 0.01, 0.02, 0.05\}$ for RAPS. With $\beta = 1$ fixed, we search $M \in \{5, 6, 7, 8, 9, 10\}$ and $\gamma \in \{1.0, 10^{-1}, 10^{-2}, 10^{-3}\}$. We then tune FPS, initialized at zero $\mathbf{a}^0$, using AdamW (weight decay $10^{-4}$). The final hyperparameters are as follows. For $\alpha = 0.05$: APS+ours uses $M = 9$, $\gamma = 10^{-2}$; RAPS+ours uses $M = 6$, $\gamma = 10^{-2}$. For $\alpha = 0.10$: APS+ours uses $M = 7$, $\gamma = 10^{-1}$; RAPS+ours uses $M = 6$, $\gamma = 10^{-1}$. We repeat each experiment 100 times and report the mean size with its standard error, and the mean coverage.

### A.4 IMPLEMENTATION DETAILS OF TABLE 4 AND 5

The sensitivity analysis is conducted on ImageNet. We tune FPS with the base model ResNet152 with $\beta \in \{10, 1.0, 0.1, 0.01\}$, $M \in \{1, 2, 3, 4\}$, and $\gamma \in \{10^{-2}, 10^{-3}, 10^{-4}, 10^{-5}\}$ for both nominal coverage levels $\alpha \in \{0.05, 0.10\}$. All other configurations are the same as in Appendix A.1.

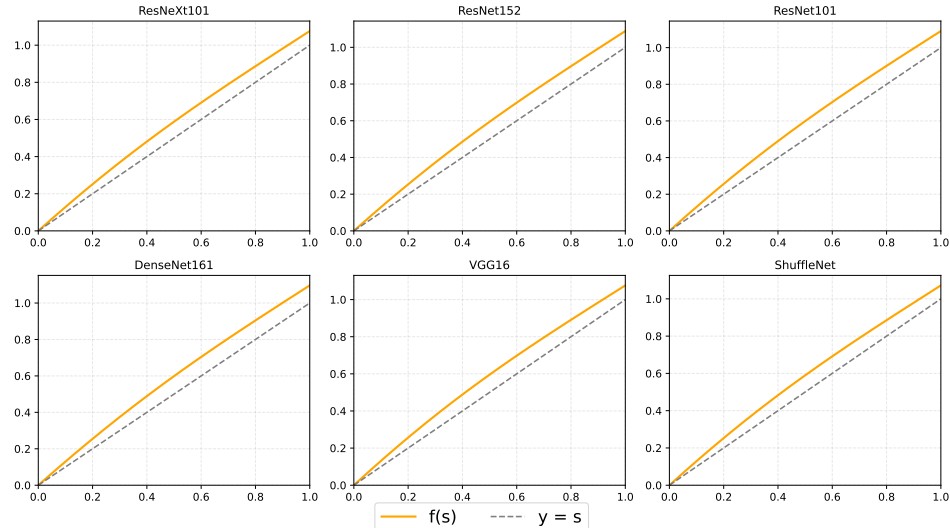

Figure 2: Learned order-preserving FPS transformation $f(s)$ (orange) across base classifiers, compared with the identity $y = s$ (gray dashed).

## B  VISUALIZATION OF THE FPS TRANSFORMATION

At the significance level $\alpha = 0.10$, we tune the FPS order-preserving transformation $f(s)$ for APS and visualize the learned $f(s)$ in Fig. 2. Unless otherwise noted, all experimental settings follow Appendix A.1. Notably, the learned transformation exhibits a highly similar qualitative pattern across all tested backbones; accordingly, FPS demonstrates robustness with respect to the base classifiers and offers general applicability to conformal prediction classification tasks.

## C  LOSS UPDATE AFTER RECALIBRATION

Intuitively, the concentration of $\boldsymbol{s}^c$ on its leading entries arises because our base classifier attains high top-$k$ accuracy, which makes the predicted probabilities sharply peaked and thus concentrated on the first few entries. Empirically, on ImageNet we consider APS with its FPS-tuned counterpart. We use ResNet-152, set $M = 1$, $\beta = 1$, and coverage levels to $\alpha \in \{0.05, 0.10\}$. We tune FPS with zero initialization using AdamW (learning rate $\gamma = 10^{-3}$, weight decay $10^{-4}$). We examine examples of $\boldsymbol{s}^c$, the average vector length, and the quantity $\sum_t \delta_t$ introduced in Assumption 2.

In the vast majority of cases, $\boldsymbol{s}^c$ is highly concentrated on a single entry, e.g., $[1.]$. In other common cases, the mass is still dominated by the first two or three entries, e.g., $[0.999, 0.0007]$, $[0.994, 0.005]$, or $[0.991, 0.004, 0.003]$. Less frequently, we observe longer tails, such as $[0.998, 0.0011, 0.00017, 0.00016, 0.00010, 0.000051, 0.000021, 0.0000082]$, and only rarely a pattern like $[0.517, 0.410]$. Overall, these patterns indicate a sharply peaked predictive distribution consistent with a high top-$k$ accuracy base classifier, supporting our concentration assumption for $\boldsymbol{s}^c$. The average length of $\boldsymbol{s}^c$ remains below 2 across $T = 50$ iterations, which is small and further supports our concentration assumption on $\boldsymbol{s}^c$. Finally, we directly report the cumulative loss update $\sum_{t=1}^{T} \delta_t$. Figure 3 reports the cumulative loss update over the first $T = 50$ iterations, computed solely from changes in $\boldsymbol{s}^c$ and excluding the gradient–descent term. The trajectories increase with diminishing increments and remain small in magnitude; empirically, they exhibit sublinear growth, i.e., $\sum_{t=1}^{T} \delta_t = o(T)$, consistent with the vanishing-loss behavior in Assumption 2.

## D  OVERLAPPING DATA SPLITS INDUCE LIMITED TUNING BIAS

In Algorithm 2, we use a hold-out set $\mathcal{D}_{\text{tune}}$ to learn the FPS transformation and then integrate it into split conformal prediction. This may raise concerns about requiring too much additional

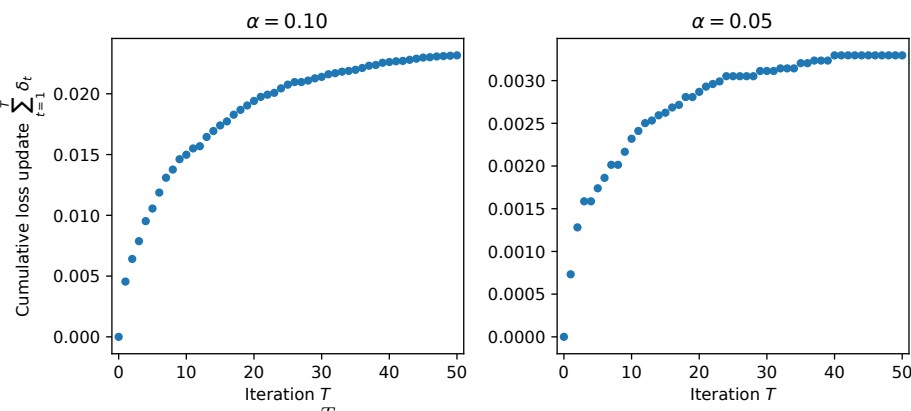

Figure 3: Cumulative loss update $\sum_{t=1}^{T} \delta_t$ over the first $T = 50$ iterations, based solely on changes in $s^c$ (gradient–descent term excluded), for $\alpha = 0.10$ (left) and $\alpha = 0.05$ (right).

Table 6: Coverage and size on ImageNet under *partially overlapping* FPS tuning and conformal calibration sets, across $\alpha$ levels and base image classifiers. APS and RAPS are baselines; +ours denotes applying our FPS framework (APS+ours, RAPS+ours).

| Model | $\alpha$ | Coverage | | | | Size | | | |
|---|---|---|---|---|---|---|---|---|---|
| | | APS | APS+ours | RAPS | RAPS+ours | APS | APS+ours | RAPS | RAPS+ours |
| ResNeXt101 | 0.05 | 0.951 | 0.951 | 0.951 | 0.951 | $20.672 \pm 0.203$ | $\mathbf{10.847} \pm 0.266$ | $3.763 \pm 0.040$ | $\mathbf{3.746} \pm 0.040$ |
| | 0.10 | 0.900 | 0.900 | 0.901 | 0.902 | $7.112 \pm 0.085$ | $\mathbf{2.894} \pm 0.042$ | $2.030 \pm 0.004$ | $\mathbf{1.994} \pm 0.003$ |
| ResNet152 | 0.05 | 0.950 | 0.951 | 0.950 | 0.950 | $14.384 \pm 0.166$ | $\mathbf{8.159} \pm 0.136$ | $4.091 \pm 0.063$ | $\mathbf{4.067} \pm 0.046$ |
| | 0.10 | 0.901 | 0.901 | 0.900 | 0.901 | $6.229 \pm 0.032$ | $\mathbf{2.975} \pm 0.033$ | $2.254 \pm 0.007$ | $\mathbf{2.182} \pm 0.006$ |
| ResNet101 | 0.05 | 0.950 | 0.950 | 0.950 | 0.949 | $15.678 \pm 0.163$ | $\mathbf{8.989} \pm 0.121$ | $4.445 \pm 0.029$ | $\mathbf{4.359} \pm 0.025$ |
| | 0.10 | 0.902 | 0.901 | 0.901 | 0.901 | $6.924 \pm 0.043$ | $\mathbf{3.259} \pm 0.034$ | $2.376 \pm 0.006$ | $\mathbf{2.298} \pm 0.003$ |
| DenseNet161 | 0.05 | 0.951 | 0.951 | 0.950 | 0.950 | $17.256 \pm 0.151$ | $\mathbf{9.980} \pm 0.146$ | $4.673 \pm 0.028$ | $\mathbf{4.671} \pm 0.019$ |
| | 0.10 | 0.900 | 0.900 | 0.900 | 0.901 | $6.759 \pm 0.073$ | $\mathbf{3.300} \pm 0.017$ | $2.353 \pm 0.005$ | $\mathbf{2.296} \pm 0.006$ |
| VGG16 | 0.05 | 0.952 | 0.951 | 0.951 | 0.951 | $24.374 \pm 0.214$ | $\mathbf{15.321} \pm 0.152$ | $8.341 \pm 0.068$ | $\mathbf{8.186} \pm 0.068$ |
| | 0.10 | 0.899 | 0.899 | 0.899 | 0.900 | $11.596 \pm 0.081$ | $\mathbf{5.735} \pm 0.045$ | $3.802 \pm 0.029$ | $\mathbf{3.628} \pm 0.022$ |
| ShuffleNet | 0.05 | 0.951 | 0.952 | 0.951 | 0.951 | $55.364 \pm 0.316$ | $\mathbf{32.924} \pm 4.682$ | $14.811 \pm 0.203$ | $\mathbf{14.365} \pm 0.179$ |
| | 0.10 | 0.898 | 0.899 | 0.901 | 0.901 | $22.990 \pm 0.226$ | $\mathbf{8.786} \pm 0.106$ | $4.968 \pm 0.040$ | $\mathbf{4.878} \pm 0.047$ |

data. However, Zeng et al. (2025) show that prediction-set coverage remains near nominal despite violations of exchangeability in the non-conformity scores, as long as the calibration set is large. In this section, we therefore allow $\mathcal{D}_{\text{tune}} := \mathcal{D} \cup \mathcal{D}^c$ and $\mathcal{D}_{\text{cal}}$ to partially overlap. Specifically, we reuse the same calibration set for tuning the FPS transformation and for the subsequent split conformal prediction, i.e., $\mathcal{D}^c := \mathcal{D}_{\text{cal}}$. We report experimental results under this data-splitting scheme. We split the dataset in a 1:2:2 ratio into a tuning set (used exclusively for gradient learning), a calibration set (shared for estimating $s^c$ and executing the split CP procedures), and an evaluation set to evaluate Coverage and Size. For example, we split the 50k-sample ImageNet validation set into three partially overlapping subsets: 10k for tuning the FPS transformation (for gradient-based learning only), 20k for both conformal calibration and searching for the threshold and the associated calibration components needed to learn the FPS transformation, and 20k for evaluating prediction set size and coverage. Apart from the changes noted above, all parameter settings are identical to those in Appendix A.

Tables 6, 7, and 8 show that even when the FPS transformation tuning set partially overlaps with the conformal calibration set, FPS attains target coverage and yields smaller prediction sets for both APS and RAPS. The improvements are consistent across $\alpha$ levels, base architectures, and multimodal datasets. In conclusion, although this data-split scheme breaks exchangeability and thus invalidates the coverage guarantee in Theorem 1, our experiments show that coverage is still achieved in practice, thereby alleviating potential concerns about data waste.

Table 7: Coverage and Size results on ImageNet-V2 under *partially overlapping* FPS tuning and conformal calibration sets, across $\alpha$ levels and base image classifiers.

| Model | $\alpha$ | Coverage | | | | Size | | | |
| --- | --- | --- | --- | --- | --- | --- | --- | --- | --- |
| | | APS | APS+ours | RAPS | RAPS+ours | APS | APS+ours | RAPS | RAPS+ours |
| ResNeXt101 | 0.05 | 0.949 | 0.950 | 0.952 | 0.951 | $69.305 \pm 0.372$ | $\mathbf{48.459} \pm 0.189$ | $18.774 \pm 0.248$ | $\mathbf{18.087} \pm 0.137$ |
| | 0.10 | 0.899 | 0.901 | 0.897 | 0.897 | $25.773 \pm 0.172$ | $\mathbf{14.067} \pm 0.068$ | $6.150 \pm 0.075$ | $\mathbf{5.723} \pm 0.048$ |
| ResNet152 | 0.05 | 0.950 | 0.951 | 0.951 | 0.951 | $42.697 \pm 0.172$ | $\mathbf{35.830} \pm 0.141$ | $15.850 \pm 0.251$ | $\mathbf{15.100} \pm 0.165$ |
| | 0.10 | 0.900 | 0.900 | 0.898 | 0.899 | $17.912 \pm 0.145$ | $\mathbf{12.439} \pm 0.083$ | $5.437 \pm 0.031$ | $\mathbf{5.410} \pm 0.022$ |
| ResNet101 | 0.05 | 0.950 | 0.950 | 0.949 | 0.950 | $49.713 \pm 0.426$ | $\mathbf{41.037} \pm 0.295$ | $19.823 \pm 0.227$ | $\mathbf{19.361} \pm 0.248$ |
| | 0.10 | 0.900 | 0.903 | 0.903 | 0.901 | $20.763 \pm 0.125$ | $\mathbf{14.329} \pm 0.060$ | $6.860 \pm 0.070$ | $\mathbf{6.380} \pm 0.023$ |
| DenseNet161 | 0.05 | 0.953 | 0.953 | 0.950 | 0.949 | $58.149 \pm 0.469$ | $\mathbf{45.249} \pm 0.257$ | $19.268 \pm 0.112$ | $\mathbf{18.487} \pm 0.073$ |
| | 0.10 | 0.897 | 0.898 | 0.904 | 0.903 | $20.267 \pm 0.156$ | $\mathbf{13.468} \pm 0.066$ | $6.626 \pm 0.077$ | $\mathbf{6.344} \pm 0.040$ |
| VGG16 | 0.05 | 0.948 | 0.947 | 0.951 | 0.950 | $56.276 \pm 0.213$ | $\mathbf{50.023} \pm 0.230$ | $30.646 \pm 0.545$ | $\mathbf{28.279} \pm 0.358$ |
| | 0.10 | 0.899 | 0.898 | 0.898 | 0.899 | $27.661 \pm 0.191$ | $\mathbf{21.459} \pm 0.125$ | $11.440 \pm 0.105$ | $\mathbf{11.124} \pm 0.073$ |
| ShuffleNet | 0.05 | 0.951 | 0.952 | 0.949 | 0.947 | $132.295 \pm 0.288$ | $\mathbf{114.352} \pm 0.266$ | $72.303 \pm 0.836$ | $\mathbf{69.546} \pm 0.585$ |
| | 0.10 | 0.903 | 0.901 | 0.903 | 0.901 | $60.811 \pm 0.436$ | $\mathbf{38.840} \pm 0.223$ | $24.045 \pm 0.323$ | $\mathbf{22.719} \pm 0.199$ |

Table 8: Coverage and Size results on Banking77 under *partially overlapping* FPS tuning and conformal calibration sets, across $\alpha$ levels and base text classifiers.

| Model | $\alpha$ | Coverage | | | | Size | | | |
| --- | --- | --- | --- | --- | --- | --- | --- | --- | --- |
| | | APS | APS+ours | RAPS | RAPS+ours | APS | APS+ours | RAPS | RAPS+ours |
| BERT | 0.05 | 0.950 | 0.950 | 0.948 | 0.949 | $2.531 \pm 0.024$ | $\mathbf{1.469} \pm 0.014$ | $1.438 \pm 0.009$ | $\mathbf{1.351} \pm 0.010$ |
| | 0.10 | 0.900 | 0.899 | 0.900 | 0.900 | $1.537 \pm 0.011$ | $\mathbf{1.155} \pm 0.130$ | $1.132 \pm 0.005$ | $\mathbf{0.976} \pm 0.001$ |
| RoBERTa | 0.05 | 0.950 | 0.950 | 0.949 | 0.949 | $2.019 \pm 0.022$ | $\mathbf{1.260} \pm 0.010$ | $1.224 \pm 0.005$ | $\mathbf{1.171} \pm 0.004$ |
| | 0.10 | 0.901 | 0.901 | 0.900 | 0.899 | $1.335 \pm 0.009$ | $\mathbf{0.969} \pm 0.001$ | $1.059 \pm 0.004$ | $\mathbf{0.968} \pm 0.001$ |
| DistilBERT | 0.05 | 0.950 | 0.950 | 0.950 | 0.951 | $2.144 \pm 0.018$ | $\mathbf{1.420} \pm 0.010$ | $1.446 \pm 0.009$ | $\mathbf{1.346} \pm 0.008$ |
| | 0.10 | 0.902 | 0.901 | 0.898 | 0.900 | $1.478 \pm 0.007$ | $\mathbf{0.979} \pm 0.001$ | $1.159 \pm 0.005$ | $\mathbf{0.979} \pm 0.001$ |
| DistilRoBERTa | 0.05 | 0.949 | 0.949 | 0.948 | 0.949 | $4.201 \pm 0.028$ | $\mathbf{2.054} \pm 0.014$ | $2.249 \pm 0.019$ | $\mathbf{1.819} \pm 0.009$ |
| | 0.10 | 0.901 | 0.902 | 0.900 | 0.902 | $2.735 \pm 0.018$ | $\mathbf{1.451} \pm 0.077$ | $1.669 \pm 0.010$ | $\mathbf{1.081} \pm 0.010$ |

# E   PROOFS

In this section, we present the proofs of Theorems 1, 2, and 3, together with Lemma 1.

## E.1   PROOF OF THEOREM 1

*Proof.* Let $E_g$ be the measurable non-conformity score in Eq. (2), and let $Z_i = (x_i, y_i, u_i)$, $i = 1, \ldots, n+1$, be i.i.d. Algorithm 2 produces $g_{\hat{a}}$ from an independent tuning set $\mathcal{D}_{\text{tune}}$. Conditioning on $\mathcal{D}_{\text{tune}}$ makes $g_{\hat{a}} \in \mathcal{G} \subseteq \mathcal{F}$ a fixed measurable map, so $E_{g_{\hat{a}}}$ is fixed and measurable as well, while $Z_1, \ldots, Z_{n+1}$ remain i.i.d. Applying the same fixed map to i.i.d. variables preserves i.i.d., hence $E_{g_{\hat{a}}}(Z_1), \ldots, E_{g_{\hat{a}}}(Z_{n+1})$ are i.i.d. given $\mathcal{D}_{\text{tune}}$. Unconditioning preserves exchangeability. Thus selecting $g_{\hat{a}}$ via the independent tuning set does not affect the i.i.d. property of the scores $\{E_{g_{\hat{a}}}(Z_i)\}_{i=1}^{n+1}$. The coverage guarantee

$$ 1 - \alpha \;\leq\; \mathbb{P}\big\{ y_{n+1} \in \mathcal{C}_{g_{\hat{a}}}(x_{n+1}, u_{n+1}, \tau) \big\} \;\leq\; 1 - \alpha + \frac{1}{n+1} $$

then follows directly from Theorem 1 of Romano et al. (2020). $\qquad\square$

## E.2   PROOF OF LEMMA 1

*Proof.* For some $\varepsilon > 0$, define the $\varepsilon$-lifted function

$$ \tilde{f}'_\varepsilon(s) = f^{\star\prime}(s) + \varepsilon, \qquad \tilde{f}_\varepsilon(0) = 0. $$

Then $\tilde{f}_\varepsilon \in C([0, D])$ is strictly increasing and $\|\tilde{f}_\varepsilon - f^\star\|_\infty \leq \varepsilon D$.

Set $\psi_\varepsilon(s) = \log\left(\tilde{f}'_\varepsilon(s)\right) = \log\left(f^{\star\,\prime}(s) + \varepsilon\right) \in C([0, D])$. By the Stone-Weierstrass theorem (Stone, 1948; Rudin, 1987), the algebra generated by $\{1, \sin(ms), \cos(ms) : m \geq 1\}$ is uniformly dense in $C([0, 2\pi])$. Via the linear change of variable $\theta = \frac{2\pi}{D}s$, this density transfers to $C([0, D])$. Hence for $\psi_\varepsilon \in C([0, D])$, there exist trigonometric polynomials

$$\phi_M(s) = a_0 + \sum_{m=1}^{M} \left(a_{2m-1}\sin(ms) + a_{2m}\cos(ms)\right) \tag{4}$$

such that $\|\phi_M - \psi_\varepsilon\|_\infty \to 0$ as $M \to \infty$.

Define $g_M$ by $g'_M(s) = \exp\left(\phi_M(s)\right)$ and $g_M(0) = 0$. Since $\psi_\varepsilon(s)$ is continuous and bounded on $[0, D]$; denote $B_\varepsilon := \|\psi_\varepsilon\|_\infty < \infty$. Fix any $\eta \in (0, 1]$. Because $\|\phi_M - \psi_\varepsilon\|_\infty \to 0$, there exists $M_0$ such that for all $M \geq M_0$,

$$\|\phi_M - \psi_\varepsilon\|_\infty \leq \eta \quad \Rightarrow \quad \max\{\phi_M(s), \psi_\varepsilon(s)\} \leq B_\varepsilon + \eta \text{ for all } s \in [0, D].$$

By the mean value theorem, for each $s \in [0, D]$ there exists $\xi(s)$ between $\phi_M(s)$ and $\psi_\varepsilon(s)$ such that

$$\left|e^{\phi_M(s)} - e^{\psi_\varepsilon(s)}\right| = e^{\xi(s)}\left|\phi_M(s) - \psi_\varepsilon(s)\right| \leq e^{B_\varepsilon+\eta}\left|\phi_M(s) - \psi_\varepsilon(s)\right|.$$

Hence, for all $M \geq M_0$ and all $s \in [0, D]$,

$$|g'_M(s) - \tilde{f}'_\varepsilon(s)| = \left|e^{\phi_M(s)} - e^{\psi_\varepsilon(s)}\right| \leq e^{B_\varepsilon+\eta}\left|\phi_M(s) - \psi_\varepsilon(s)\right|.$$

Integrating the pointwise bound from $0$ to $s$ and taking the supremum over $s \in [0, D]$ gives

$$\|g_M - \tilde{f}_\varepsilon\|_\infty \leq \int_0^D e^{B_\varepsilon+\eta}|\phi_M(t) - \psi_\varepsilon(t)|\,dt \leq D\,e^{B_\varepsilon+\eta}\|\phi_M - \psi_\varepsilon\|_\infty.$$

Therefore,

$$\inf_{g \in \mathcal{G}}\|g - f^\star\|_\infty \leq \|g_M - f^\star\|_\infty$$

$$\leq \|g_M - \tilde{f}_\varepsilon\|_\infty + \|\tilde{f}_\varepsilon - f^\star\|_\infty$$

$$\leq D\left(e^{B_\varepsilon+\eta}\|\phi_M - \psi_\varepsilon\|_\infty + \varepsilon\right).$$

Since $\varepsilon > 0$ is arbitrary, letting $\varepsilon \downarrow 0$ yields

$$\lim_{M \to \infty} \inf_{g \in \mathcal{G}}\|g - f^\star\|_\infty = 0,$$

which completes the proof. $\qquad\qquad\square$

### E.3 Proof of Theorem 2

*Proof.* We divide the proof into three steps.

**Step 1 ($\mathcal{L}(\cdot)$ is Lipschitz in function $g$).** Denote one sample class-probability vector as $Z = (\{s_i^c\}_{i=1}^q, \{s_i\}_{i=1}^K)$ and the corresponding loss:

$$\ell(g; Z) := \sum_{k=1}^{K} \sigma\left(\beta^{-1}\left\{\sum_{i=1}^{q} g(s_i^c) - \sum_{i=1}^{k} g(s_i)\right\}\right).$$

Since the sigmoid function is $\frac{1}{4}$-Lipschitz, for any $g_1, g_2 \in \mathcal{G}$,

$$|\ell(g_1; Z) - \ell(g_2; Z)| \leq \frac{1}{4\beta}\sum_{k=1}^{K}\left|\sum_{i=1}^{q}(g_1 - g_2)(s_i^c) - \sum_{i=1}^{k}(g_1 - g_2)(s_i)\right|$$

$$\leq \frac{1}{4\beta}\sum_{k=1}^{K}\left(\sum_{i=1}^{q}|g_1 - g_2|(s_i^c) + \sum_{i=1}^{k}|g_1 - g_2|(s_i)\right)$$

$$\leq \frac{Kq + \frac{K(K+1)}{2}}{4\beta}\|g_1 - g_2\|_\infty.$$

Since $q$ may vary across iterations, we upper bound it by $K$. Finally, taking expectations yields the Lipschitz property:

$$|\mathcal{L}(g_1) - \mathcal{L}(g_2)| \leq L_K \|g_1 - g_2\|_\infty, \quad L_K = \frac{K(3K+1)}{8\beta}. \tag{5}$$

**Step 2 (Rademacher complexity).** Let $\widehat{\mathfrak{R}}_n(\mathcal{H})$ denote the empirical Rademacher complexity. Note that $\ell(\cdot)$ is $L_K$-Lipschitz by Eq. (5), by the vector contraction inequality (Ledoux & Talagrand, 2013; Maurer, 2016),

$$\widehat{\mathfrak{R}}_n(\ell \circ \mathcal{G}) \leq L_K \, \widehat{\mathfrak{R}}_n(\mathcal{G}; \|\cdot\|_\infty).$$

For $g_{\mathbf{a}}(s) = \int_0^s \exp(\phi_{\mathbf{a}}(t))dt$ with $\phi_{\mathbf{a}}(\cdot)$ defined in Eq. (4) and $\|\mathbf{a}\|_1 \leq A$, we have $|g_{\mathbf{a}}(s)| \leq s\,e^A \leq D\,e^A$ and, moreover, $|g_{\mathbf{a}_1}(s) - g_{\mathbf{a}_2}(s)| \leq D\,e^A \|\mathbf{a}_1 - \mathbf{a}_2\|_1$. Hence by standard Dudley bounds:

$$\widehat{\mathfrak{R}}_n(\mathcal{G}; \|\cdot\|_\infty) \lesssim \frac{D\,e^A}{\sqrt{n}}.$$

Finally, with probability exceeding $1 - \delta$,

$$\sup_{g \in \mathcal{G}} |\mathcal{L}(g) - \mathcal{L}_n(g)| \lesssim L_K \frac{D\,e^A}{\sqrt{n}} + \sqrt{\frac{\log(1/\delta)}{n}}. \tag{6}$$

**Step 3: Generalization bound.** Decompose

$$\mathcal{L}(g_{\widehat{\mathbf{a}}}) - \mathcal{L}(f^\star) = \big[\mathcal{L}(g_{\widehat{\mathbf{a}}}) - \mathcal{L}_n(g_{\widehat{\mathbf{a}}})\big] + \big[\mathcal{L}_n(g_{\widehat{\mathbf{a}}}) - \inf_{g \in \mathcal{G}} \mathcal{L}_n(g)\big]$$

$$+ \big[\inf_{g \in \mathcal{G}} \mathcal{L}_n(g) - \inf_{g \in \mathcal{G}} \mathcal{L}(g)\big] + \big[\inf_{g \in \mathcal{G}} \mathcal{L}(g) - \mathcal{L}(f^\star)\big],$$

where the second term is bounded by $\varepsilon_{\mathrm{opt}}$ with Assumption 1, and the first and third brackets are each bounded by the uniform deviation in (6). For the fourth term, pick $\tilde{g} \in \mathcal{G}$ with $\|\tilde{g} - f^\star\|_\infty \leq \delta_M$ (cf. Lemma 1); then by the Lipschitz property (5),

$$0 \leq \inf_{g \in \mathcal{G}} \mathcal{L}(g) - \mathcal{L}(f^\star) \leq \mathcal{L}(\tilde{g}) - \mathcal{L}(f^\star) \leq L_K \, \delta_M.$$

Collecting the bounds yields

$$L(g_{\widehat{\mathbf{a}}}) - L(f^\star) \leq C_1 L_K \frac{D\,e^A}{\sqrt{n}} + C_2 \sqrt{\frac{\log(1/\delta)}{n}} + L_K \delta_M + \varepsilon_{\mathrm{opt}},$$

with probability at least $1 - \delta$, where $C_1, C_2 > 0$ are universal constants. Finally, by Lemma 1, $\delta_M \to 0$ as $M \to \infty$, and thus $\mathcal{L}(g_{\widehat{\mathbf{a}}}) - \mathcal{L}(f^\star) = \varepsilon_{\mathrm{opt}} + o_{\mathbb{P}}(1)$ as stated. $\qquad\square$

### E.4 PROOF OF THEOREM 3

*Proof.* We divide the proof into three steps.

**Step 1 (L-smooth in a).** Fix $t \geq 1$ and freeze the calibration $s^{c,t-1}$. Consider the map $\mathbf{a} \mapsto \mathcal{L}_n(g_{\mathbf{a}}, s^{c,t-1})$ for parameters constrained by $\|\mathbf{a}\|_1 \leq A$. Write $\phi_{\mathbf{a}}(u) = \langle \mathbf{a}, \mathbf{b}(u) \rangle$ with

$$\mathbf{b}(u) = (1, \sin u, \cos u, \ldots, \sin Mu, \cos Mu)^\top$$

so that $g_{\mathbf{a}}(s) = \int_0^s e^{\phi_{\mathbf{a}}(u)} \, du$. Define $J_{\mathbf{a}}(s) := \nabla_{\mathbf{a}} g_{\mathbf{a}}(s) = \int_0^s e^{\phi_{\mathbf{a}}(u)} \mathbf{b}(u) \, du$, and the inner term $z_{i,k}(\mathbf{a}) = \beta^{-1}\big\{\sum_{j=1}^q g_{\mathbf{a}}(s_j^{c,t-1}) - \sum_{j=1}^k g_{\mathbf{a}}(s_{i,j})\big\}$. Then, the gradient w.r.t. $\mathbf{a}$ is

$$\nabla_{\mathbf{a}} \mathcal{L}_n(g_{\mathbf{a}}, s^{c,t-1}) = \frac{1}{\beta n} \sum_{i=1}^n \sum_{k=1}^K \sigma'(z_{i,k}(\mathbf{a})) \Big(\sum_{j=1}^q J_{\mathbf{a}}(s_j^{c,t-1}) - \sum_{j=1}^k J_{\mathbf{a}}(s_{i,j})\Big).$$

For two parameters $\mathbf{a}_1, \mathbf{a}_2$, by $0 < \sigma'(x) \leq 1/4$,

$$\big\|\nabla_{\mathbf{a}} \mathcal{L}_n(g_{\mathbf{a}_1}, s^{c,t-1}) - \nabla_{\mathbf{a}} \mathcal{L}_n(g_{\mathbf{a}_2}, s^{c,t-1})\big\| \leq \frac{1}{\beta} \sum_{k=1}^K \Big\{|\Delta\sigma'_{i,k}| \cdot \big\|\sum J_{\mathbf{a}_1}\big\| + \tfrac{1}{4}\big\|\sum(J_{\mathbf{a}_1} - J_{\mathbf{a}_2})\big\|\Big\}.$$

Here we set, for each sample $i$ and level $k$,

$$\Delta\sigma'_{i,k} := \sigma'\big(z_{i,k}(\mathbf{a}_1)\big) - \sigma'\big(z_{i,k}(\mathbf{a}_2)\big), \qquad \sum J_{\mathbf{a}} := \sum_{j=1}^{q} J_{\mathbf{a}}(s_j^{c,t-1}) - \sum_{j=1}^{k} J_{\mathbf{a}}(s_{i,j}).$$

Since $\sigma'$ is $L_{\sigma'}$-Lipschitz with $L_{\sigma'} = \sup_x |\sigma''(x)| \leq 1/6\sqrt{3} \leq 1/4$ and $\big\| \sum J_{\mathbf{a}_1} \big\| \leq (q+k)De^A$, we get

$$|\Delta\sigma'_{i,k}| \leq L_{\sigma'} |z_{i,k}(\mathbf{a}_1) - z_{i,k}(\mathbf{a}_2)| \leq \frac{1}{4\beta}(q+k)\, De^A \, \|\mathbf{a}_1 - \mathbf{a}_2\|_1.$$

Also, by the mean-value argument in parameter space,

$$\big\| \sum(J_{\mathbf{a}_1} - J_{\mathbf{a}_2}) \big\| \leq (q+k)De^A \, \|\mathbf{a}_1 - \mathbf{a}_2\|_1.$$

Combining and summing over $k$ (using $q \leq K$) yields

$$\big\| \nabla_{\mathbf{a}}\mathcal{L}_n(g_{\mathbf{a}_1}, \boldsymbol{s}^{c,t-1}) - \nabla_{\mathbf{a}}\mathcal{L}_n(g_{\mathbf{a}_2}, \boldsymbol{s}^{c,t-1}) \big\| \leq L_A \, \|\mathbf{a}_1 - \mathbf{a}_2\|_1,$$

where $L_A = \frac{K\, De^A}{24\,\beta}\left[ \frac{De^A}{\beta}\left(14K^2 + 9K + 1\right) + 3(3K+1) \right].$

**Step 2 (One-step descent).** With step size $\gamma \in (0, 1/L_A]$, the descent lemma (Beck, 2017) gives

$$\mathcal{L}_n\big(g_{\mathbf{a}^t}, \boldsymbol{s}^{c,t-1}\big) \leq \mathcal{L}_n\big(g_{\mathbf{a}^{t-1}}, \boldsymbol{s}^{c,t-1}\big) - \frac{\gamma}{2}\big\| \nabla_{\mathbf{a}}\mathcal{L}_n\big(g_{\mathbf{a}^{t-1}}, \boldsymbol{s}^{c,t-1}\big) \big\|^2.$$

**Step 3 (Limit point stationarity).** By Assumption 2,

$$\mathcal{L}_n\big(g_{\mathbf{a}^t}, \boldsymbol{s}^{c,t}\big) \leq \mathcal{L}_n\big(g_{\mathbf{a}^t}, \boldsymbol{s}^{c,t-1}\big) + \delta_t,$$

which, combined with Step 2, yields

$$\mathcal{L}_n\big(g_{\mathbf{a}^t}, \boldsymbol{s}^{c,t}\big) \leq \mathcal{L}_n\big(g_{\mathbf{a}^{t-1}}, \boldsymbol{s}^{c,t-1}\big) - \frac{\gamma}{2}\big\| \nabla_{\mathbf{a}}\mathcal{L}_n\big(g_{\mathbf{a}^{t-1}}, \boldsymbol{s}^{c,t-1}\big) \big\|^2 + \delta_t.$$

Summing over $t = 1, \ldots, T$ and using $\mathcal{L}_n \geq 0$ gives

$$\frac{1}{T}\sum_{t=1}^{T} \big\| \nabla_{\mathbf{a}}\mathcal{L}_n\big(g_{\mathbf{a}^{t-1}}, \boldsymbol{s}^{c,t-1}\big) \big\|^2 \leq \frac{2}{\gamma}\frac{\mathcal{L}_n\big(g_{\mathbf{a}^0}, \boldsymbol{s}^{c,0}\big)}{T} + \frac{2}{\gamma}\cdot\frac{1}{T}\sum_{t=1}^{T}\delta_t.$$

Letting $T \to \infty$ and invoking $\frac{1}{T}\sum_{t=1}^{T}\delta_t \to 0$ from Assumption 2 yields

$$\lim_{T\to\infty}\frac{1}{T}\sum_{t=1}^{T}\big\| \nabla_{\mathbf{a}}\mathcal{L}_n\big(g_{\mathbf{a}^t}, \boldsymbol{s}^{c,t}\big) \big\|^2 = 0, \quad \text{and hence} \quad \liminf_{t\to\infty}\big\| \nabla_{\mathbf{a}}\mathcal{L}_n\big(g_{\mathbf{a}^t}, \boldsymbol{s}^{c,t}\big) \big\| = 0,$$

which completes the proof. $\qquad\qquad\qquad\qquad\qquad\qquad\qquad\qquad\qquad\qquad\qquad\quad\square$

## F    COMPARISON WITH LEAST AMBIGUOUS SET-VALUED CLASSIFIER

The Least Ambiguous Set-valued Classifier (LAC; (Sadinle et al., 2019)) is a method known for size efficiency. LAC induces prediction sets by thresholding the model scores $\hat{\boldsymbol{\pi}}(x)$; in our notation this corresponds to the following non-conformity score and prediction rule:

$$E_{\text{LAC}}(x, y) = 1 - \hat{\pi}_y(x).$$

Given a calibration set $\mathcal{D}_{\text{cal}} = \{(x_i, y_i)\}_{i=1}^n$, compute $e_i = E_{\text{LAC}}(x_i, y_i)$ and the empirical $(1-\alpha)$ quantile

$$\tau = \inf\Big\{ e : \frac{\big|\{ i : e_i \leq e \}\big|}{n} \geq \frac{\big\lceil (n+1)(1-\alpha) \big\rceil}{n} \Big\}.$$

For a new input $x$, the LAC prediction set is

$$\mathcal{C}(x) = \big\{ y \in \mathcal{Y} : E_{\text{LAC}}(x, y) \leq \tau \big\} = \big\{ y \in \mathcal{Y} : \hat{\pi}_y(x) \geq 1 - \tau \big\}.$$

Table 9: SSCV (mean $\pm$ SE) results on ImageNet.

| Model | $\alpha = 0.05$ | | | $\alpha = 0.10$ | | |
|---|---|---|---|---|---|---|
| | APS+ours | RAPS+ours | LAC | APS+ours | RAPS+ours | LAC |
| ResNeXt101 | $0.031 \pm 0.001$ | $0.030 \pm 0.003$ | $0.082 \pm 0.004$ | $0.057 \pm 0.003$ | $0.068 \pm 0.008$ | $0.347 \pm 0.063$ |
| ResNet152 | $0.026 \pm 0.001$ | $0.032 \pm 0.004$ | $0.118 \pm 0.005$ | $0.042 \pm 0.003$ | $0.083 \pm 0.012$ | $0.246 \pm 0.014$ |
| ResNet101 | $0.030 \pm 0.002$ | $0.033 \pm 0.004$ | $0.107 \pm 0.004$ | $0.054 \pm 0.003$ | $0.082 \pm 0.010$ | $0.202 \pm 0.011$ |
| DenseNet161 | $0.026 \pm 0.001$ | $0.038 \pm 0.007$ | $0.087 \pm 0.004$ | $0.045 \pm 0.004$ | $0.061 \pm 0.009$ | $0.264 \pm 0.071$ |
| VGG16 | $0.020 \pm 0.001$ | $0.026 \pm 0.004$ | $0.064 \pm 0.004$ | $0.033 \pm 0.002$ | $0.040 \pm 0.007$ | $0.223 \pm 0.008$ |
| ShuffleNet | $0.022 \pm 0.001$ | $0.028 \pm 0.006$ | $0.130 \pm 0.005$ | $0.031 \pm 0.001$ | $0.033 \pm 0.003$ | $0.171 \pm 0.003$ |

As in standard split conformal classification, optional tie-breaking randomization can be used on the boundary $E_{\text{LAC}}(x, y) = \tau$ to ensure exact finite-sample coverage.

Though LAC is highly size efficient, it sacrifices (group) conditional coverage such as coverage conditioned on the realized set size $|\mathcal{C}(X)|$. To assess such heterogeneity in coverage, a common metric is the Size-Stratified Coverage Violation (SSCV; (Angelopoulos et al., 2021)), which measures deviations from the target level $1 - \alpha$ across strata defined by set size; we formalize SSCV below.

Let $\mathcal{D}_{\text{eval}} = \{(x_i, y_i)\}_{i=1}^{n_{\text{eval}}}$ be an evaluation set. Fix a partition of possible set sizes $\{S_j\}_{j=1}^s$ with $\bigcup_{j=1}^s S_j = \{1, \ldots, K\}$ and $S_j \cap S_{j'} = \emptyset$ for $j \neq j'$. Define the index sets

$$\mathcal{J}_j = \big\{ i \in \{1, \ldots, n_{\text{eval}}\} : \big|\mathcal{C}(x_i)\big| \in S_j \big\}.$$

Then the empirical SSCV at miscoverage $\alpha$ is

$$\widehat{\text{SSCV}}_\alpha\big(\mathcal{C}, \{S_j\}_{j=1}^s\big) = \sup_{j : |\mathcal{J}_j| > 0} \left| \frac{1}{|\mathcal{J}_j|} \sum_{i \in \mathcal{J}_j} \mathbb{1}\{ y_i \in \mathcal{C}(x_i) \} - (1 - \alpha) \right|.$$

We split the 50k-sample ImageNet validation set into three disjoint subsets: 20k for tuning the FPS transformation (split evenly into 10k for gradient-based learning and 10k for searching the threshold and associated calibration components), 10k for conformal calibration, and 20k for evaluating SSCV. Following Angelopoulos et al. (2021), we select $k_{\text{reg}}$ and search $\lambda \in \{0.00001, 0.0001, 0.0008, 0.001, 0.0015, 0.002\}$ for RAPS. We further set $M = 1$ and $\beta = 1$, and tune FPS initialized at zero $\mathbf{a}^0$, with AdamW (learning rate $\gamma = 10^{-3}$, weight decay $10^{-4}$). Each experiment is repeated 10 times and we report the mean SSCV and its standard error. For SSCV, prediction sets are stratified by size into bins 0–1, 2–3, 4–10, 11–100, and 101–1000.

From Table 9, our method suffers only a minor loss in (group-)conditional coverage, as indicated by smaller SSCV, making it suitable for fairness-sensitive settings and scenarios that require adaptiveness.

## G  DISCUSSION ON TEMPERATURE SCALING

A relevant post-hoc method that also modifies a model's predictions is Temperature Scaling (Guo et al., 2017), which calibrates predicted class probabilities and yields smaller prediction sets when incorporated into conformal prediction procedures. Importantly, temperature scaling and FPS are not competitors but complementary stages in the same pipeline. One feasible approach is to first fit a single temperature parameter to improve calibration which still preserves class order; then treat the calibrated probabilities as $s(x)$ and feed them to FPS, which learns an order-preserving transformation and further reduces prediction set size. This design demonstrates FPS's potential and flexibility for enhancing conformal predictors.

## H  LARGE LANGUAGE MODEL USAGE

In accordance with the ICLR policy on Large Language Model (LLM) usage, we disclose that LLMs were used exclusively for polishing the writing. They were not employed in the development of ideas, the theoretical analysis, the design of experiments, or any other substantive aspects of this work.

