# OpenReview forum: "Efficient Conformal Prediction with Order-Preserving Predictions for Classifiers"
_ICLR.cc/2026/Conference — ICLR 2026 Conference Withdrawn Submission_

### Official Review · Reviewer_AfPW · 2025-10-14

**Soundness:** 1
**Presentation:** 2
**Contribution:** 2
**Rating:** 2
**Confidence:** 4

**Summary:**

This paper proposes an intermediate step that can be inserted into the APS or RAPS conformal procedures to reduce set sizes while maintaining a marginal coverage guarantee. This step involves applying a learned transformation of the softmax vector before applying the APS or RAPS conformal score function. This transformation is optimized on a holdout dataset to minimize expected set size.

**Strengths:**

1. The problem studied in the paper is an important one: in many practical settings, prediction sets that are extremely large are not useful, so we want to minimize set size subject to the desired coverage constraints.
2. The experiments are extensive, spanning multiple model architectures, datasets, and coverage levels.

**Weaknesses:**

**Main concern**: Although the high-level problem of optimizing prediction set size is an important one, the authors focus on a restricted version of this problem for which I am not convinced they provide a practically useful solution. There are three commonly used conformal score functions for classification: the *LAC* score corresponding to the least ambiguous set-valued classifier from Sadinle et al., 2019, *APS*, and *RAPS*. At a fixed marginal coverage level, LAC generally produces much small sets that APS or RAPS. So why would someone ever use APS or RAPS? The answer is that these score functions were created with more than marginal coverage in mind; the goal of these functions is to create prediction sets with approximate _X-conditional coverage_ --- that is,
$$P(Y \in C(X) \mid X = x) \geq 1- \alpha$$
for all $x$.  This is explained in the papers that proposed APS and RAPS.

The paper restricts itself to studying the APS and RAPS conformal score functions. This in and of itself is not a problem; it simply implies that we are in a problem setting where we care about having good conditional coverage. However, the experiments in the main paper do not evaluate the conditional coverage of the methods. The authors did perform experiments that evaluate conditional coverage, but these are hidden in Appendix F (which is never referenced in the main paper!). In Table 9 of Appendix F, we see the Size-Stratified Coverage Violation (a measure of conditional-coverage) of the proposed methods, but the values for the baselines (raw APS and raw RAPS) are not provided.

To summarize: if I can be convinced that the procedure proposed in this paper reduce APS/RAPS set sizes _without significantly harming class-conditional coverage_, I would be willing to raise my score.

**Other weaknesses**: There are some places the writing could be improved.
* “RAPS yields smaller sets with valid coverage but keeps predicted probabilities
fixed, limiting flexibility and leaving potential gains untapped” (line 48) — at this point in the introduction I don’t know what this means, and I feel as though it should either be made more vague so that I don't expect to understand it, or more specific so that I do fully understand it.
* “Thereby adhering to the posthoc principle”  (line 69) - I'm not sure what this means
* "Besides" on line 96 -- I would replace this with "In addition"
* "converges to a stationary point in the sense of limit points" (line 99) -- this reads strangely.
* “vector of predicted class-probability” (line 181) -- should be "vector of predicted class probabilities"
* Algorithm 1 should be referenced in the text.
* Set size objective equation (line 191): I believe this definition is only valid if s is assumed to be sorted, so it should be defined as such in the preceding paragraph

**Questions:**

1. I would like to see tables of SSCV and set size for APS, RAPS, and LAC with and without the proposed method. Can it be shown that the proposed method reduces set size without hurting SSCV?
2. What is the randomness in the experiments? I assume the random seeds determines the splitting of data into FPS tuning, calibration, and test sets? It may be useful to explicitly state this.

---

> ### Author Response · Authors · 2025-11-24
>
> We sincerely thank the reviewer for their detailed and insightful feedback. The comments are extremely helpful, and we are particularly grateful for the focus on conditional coverage (SSCV).
>
> We are confident that our method does indeed reduce set size without significantly harming conditional coverage.
>
> ### 1. Conditional Coverage
>
> To directly address the reviewer's request, we directly compared original APS/RAPS and APS/RAPS augmented with our FPS framework. Our full results will demonstrate exactly what the reviewer asked for: our method reduces the average set size for both APS and RAPS while achieving comparable WSC and SSCV, indicating that conditional coverage is not significantly harmed. We will also include LAC, as requested, to clearly show the tradeoff: LAC produces smaller sets but at the cost of much worse conditional coverage (lower WSC, higher SSCV), justifying the use of APS/RAPS and our method in this setting.
>
> **Table A:** Size, WSC, and SSCV on ImageNet for different conformal methods and their FPS variants with $\alpha = 0.05$ across models.
>
> | Model       | Method    | Size   | WSC   | SSCV  |
> | ----------- | --------- | ------ | ----- | ----- |
> | ResNeXt101  | APS       | 20.831 | 0.929 | 0.040 |
> | ResNeXt101  | APS+ours  | 11.066 | 0.926 | 0.033 |
> | ResNeXt101  | RAPS      | 9.826  | 0.928 | 0.029 |
> | ResNeXt101  | RAPS+ours | 7.398  | 0.926 | 0.037 |
> | ResNeXt101  | LAC       | 3.794  | 0.922 | 0.082 |
> | ResNet152   | APS       | 14.218 | 0.933 | 0.030 |
> | ResNet152   | APS+ours  | 8.735  | 0.929 | 0.024 |
> | ResNet152   | RAPS      | 11.211 | 0.932 | 0.032 |
> | ResNet152   | RAPS+ours | 7.661  | 0.929 | 0.028 |
> | ResNet152   | LAC       | 3.570  | 0.920 | 0.125 |
> | ResNet101   | APS       | 15.909 | 0.934 | 0.033 |
> | ResNet101   | APS+ours  | 9.988  | 0.931 | 0.030 |
> | ResNet101   | RAPS      | 11.891 | 0.934 | 0.035 |
> | ResNet101   | RAPS+ours | 8.419  | 0.929 | 0.041 |
> | ResNet101   | LAC       | 3.963  | 0.921 | 0.098 |
> | DenseNet161 | APS       | 17.004 | 0.930 | 0.031 |
> | DenseNet161 | APS+ours  | 10.497 | 0.926 | 0.025 |
> | DenseNet161 | RAPS      | 12.695 | 0.930 | 0.030 |
> | DenseNet161 | RAPS+ours | 9.567  | 0.926 | 0.030 |
> | DenseNet161 | LAC       | 4.213  | 0.920 | 0.120 |
> | VGG16       | APS       | 24.518 | 0.935 | 0.027 |
> | VGG16       | APS+ours  | 17.0   | 0.936 | 0.021 |
> | VGG16       | RAPS      | 16.623 | 0.937 | 0.027 |
> | VGG16       | RAPS+ours | 13.159 | 0.936 | 0.044 |
> | VGG16       | LAC       | 7.160  | 0.927 | 0.067 |
> | ShuffleNet  | APS       | 54.987 | 0.935 | 0.027 |
> | ShuffleNet  | APS+ours  | 31.794 | 0.934 | 0.018 |
> | ShuffleNet  | RAPS      | 37.935 | 0.934 | 0.035 |
> | ShuffleNet  | RAPS+ours | 26.024 | 0.933 | 0.043 |
> | ShuffleNet  | LAC       | 12.838 | 0.927 | 0.133 |
>
>
>
> **Table B:** Size, WSC, and SSCV on ImageNet for different conformal methods and their FPS variants with $\alpha = 0.10$ across models.
>
> | Model       | Method    | Size   | WSC   | SSCV  |
> | ----------- | --------- | ------ | ----- | ----- |
> | ResNeXt101  | APS       | 7.096  | 0.866 | 0.071 |
> | ResNeXt101  | APS+ours  | 4.824  | 0.863 | 0.054 |
> | ResNeXt101  | RAPS      | 4.753  | 0.864 | 0.075 |
> | ResNeXt101  | RAPS+ours | 3.746  | 0.860 | 0.080 |
> | ResNeXt101  | LAC       | 1.684  | 0.847 | 0.295 |
> | ResNet152   | APS       | 6.343  | 0.873 | 0.062 |
> | ResNet152   | APS+ours  | 4.608  | 0.870 | 0.046 |
> | ResNet152   | RAPS      | 4.561  | 0.873 | 0.076 |
> | ResNet152   | RAPS+ours | 3.727  | 0.870 | 0.075 |
> | ResNet152   | LAC       | 1.747  | 0.851 | 0.273 |
> | ResNet101   | APS       | 6.955  | 0.875 | 0.057 |
> | ResNet101   | APS+ours  | 4.979  | 0.873 | 0.053 |
> | ResNet101   | RAPS      | 4.752  | 0.875 | 0.081 |
> | ResNet101   | RAPS+ours | 3.838  | 0.873 | 0.071 |
> | ResNet101   | LAC       | 1.762  | 0.855 | 0.265 |
> | DenseNet161 | APS       | 6.910  | 0.872 | 0.067 |
> | DenseNet161 | APS+ours  | 5.134  | 0.870 | 0.055 |
> | DenseNet161 | RAPS      | 4.433  | 0.868 | 0.089 |
> | DenseNet161 | RAPS+ours | 3.835  | 0.867 | 0.079 |
> | DenseNet161 | LAC       | 1.917  | 0.847 | 0.329 |
> | VGG16       | APS       | 11.844 | 0.878 | 0.049 |
> | VGG16       | APS+ours  | 9.513  | 0.879 | 0.039 |
> | VGG16       | RAPS      | 10.215 | 0.880 | 0.035 |
> | VGG16       | RAPS+ours | 8.454  | 0.879 | 0.033 |
> | VGG16       | LAC       | 2.989  | 0.854 | 0.209 |
> | ShuffleNet  | APS       | 22.625 | 0.878 | 0.046 |
> | ShuffleNet  | APS+ours  | 16.116 | 0.878 | 0.032 |
> | ShuffleNet  | RAPS      | 15.554 | 0.878 | 0.025 |
> | ShuffleNet  | RAPS+ours | 12.225 | 0.876 | 0.027 |
> | ShuffleNet  | LAC       | 4.305  | 0.859 | 0.163 |
>
>
>
> ### 2. Randomness
>
> We use random seeds to ensure reproducible data splitting, specifically for splitting the full dataset into four parts.
>
> We are grateful for the reviewer's constructive and actionable feedback, which we believe will significantly improve the quality and impact of our paper.

---

### Official Review · Reviewer_rYPi · 2025-10-18

**Soundness:** 3
**Presentation:** 3
**Contribution:** 2
**Rating:** 4
**Confidence:** 4

**Summary:**

This paper proposed a method named Flexible Prediction Sets (FPS), a post-hoc method for improving the size-efficiency of conformal prediction (CP) in classification tasks.
The core idea is to learn a monotonic, order-preserving transformation that rescales a model's predicted class probabilities.
This transformation, learned by optimizing a smooth surrogate for the set-size objective on a tuning set, reshapes the probability magnitudes to make the true label more distinct, thereby yielding smaller prediction sets while rigorously maintaining the target coverage guarantee.

**Strengths:**

1. Novelty and Significance: The proposed method is simple, intuitive, and highly effective. It addresses the key challenge of size-efficiency in CP without requiring model retraining or complex non-conformity score design, making it a practical and widely applicable tool.
2. Theoretical Soundness: The work is well-supported by strong theoretical results, including a formal proof that the transformation preserves the finite-sample coverage guarantee of CP. The authors also provide a generalization bound for the learned function and prove the convergence of their optimization algorithm to a stationary point.
3. Empirical Validity: The experimental evaluation is comprehensive and convincing. FPS demonstrates consistent and significant reductions in prediction set size across diverse benchmarks, multiple modern classifiers, and some CP baselines (APS and RAPS). The gains are particularly notable under distribution shift, highlighting the method's robustness.

**Weaknesses:**

This paper, while presenting a well-executed empirical study, suffers from significant weaknesses in its positioning relative to prior art and the framing of its contributions.
These issues call into question the novelty and overall significance of the work.

**1. Overlap with Prior Work and Unclear Novelty**

The core contribution, a post-hoc learned adapter for improving CP efficiency, lacks novelty due to its strong resemblance to the "C-Adapter" method proposed by Liu et al. (2024).
Both FPS and C-Adapter learn a transformation on the classifier's outputs using a tuning set to minimize a surrogate of the prediction set size. The paper fails to cite or discuss this highly relevant work, which is a critical omission.

The primary discernible difference appears to be in the parameterization of the transformation: FPS uses an exponentiated trigonometric polynomial to structurally guarantee monotonicity, whereas C-Adapter often uses a small MLP with constraints on its weights. However, the conceptual gap between these two approaches is not articulated and may be marginal. Without a direct comparison, it is impossible to assess whether the specific design choice of FPS offers any substantive advantage over C-Adapter.

The authors *shall* thoroughly discuss C-Adapter and other similar methods. They need to clearly delineate their technical and conceptual contributions over this prior work. An empirical comparison with C-Adapter on at least one benchmark is essential to justify the claim of a novel contribution(if any, I will rise the rating). If the performance is similar, the authors must provide a compelling argument for why their specific parameterization or theoretical analysis is a significant advance.

**2. Insufficient Literature Review on Order-Preserving Functions**

The emphasis on "order-preserving" transformations is presented as a key principle, yet the paper fails to engage with the existing literature on this topic within deep learning. Rahimi et al. (2020) introduced "Intra Order-Preserving Functions" specifically for calibrating multi-class neural networks. Since improved calibration is a likely mechanism behind FPS's success, this work is directly relevant. The authors should have discussed whether FPS is essentially a new form of post-hoc calibration and how it relates to methods like Rahimi's.
Other related works include Rahimi et al. (2021) and Zeng et al. (2025) about the order-preserving regularization.
This oversight weakens the foundation of the work.

The related work section requires a major revision. The authors must incorporate a discussion of order-preserving functions in the context of both model calibration and conformal prediction, citing the works mentioned above and others. This will properly contextualize their contribution and clarify what is truly new about their approach. A rising rating also possible for completing related  work.

**3. Overstated Theoretical Contribution**

The paper presents three main theoretical results, but their significance is not uniform and could be seen as misleading.

*   **Trivial Coverage Guarantee:** The main result, Theorem 1 (coverage guarantee), is a direct and standard consequence of using a split-data approach in the conformal prediction framework. As long as the transformation is learned on data independent of the calibration set, the guarantee holds for *any* such function. This is a prerequisite for the method to be valid, not a novel theoretical finding.
The more incremental theoretical contributions are the generalization bound (Theorem 2) and the convergence analysis (Theorem 3) for the specific learning procedure proposed.

The authors should reframe their theoretical contributions. They should clearly state that the coverage guarantee is a known property of the split-CP framework and instead position their analysis of the surrogate loss and optimization procedure as the core theoretical novelty of the paper. This would provide a more accurate representation of their work's contribution.

[Reference]

> [1] Liu, K., Zeng, H., Huang, J., Zhuang, H., VONG, C. M., & Wei, H. (2024). C-adapter: Adapting deep classifiers for efficient conformal prediction sets.
>
> [2] Rahimi, Amir. 2021. “Learning with Limited Data and Supervision.” PhD Thesis, The Australian National University.
>
> [3] Rahimi, Amir, Amirreza Shaban, Ching-An Cheng, Richard Hartley, and Byron Boots. 2020. “Intra Order-Preserving Functions for Calibration of Multi-Class Neural Networks.” Neural Information Processing Systems 33: 13456–67.
>
> [4] Zeng, Hao, Kangdao Liu, Bingyi Jing, and Hongxin Wei. 2025. “Parametric Scaling Law of Tuning Bias in Conformal Prediction.” Paper presented at International Conference on Machine Learning. Forty-Second International Conference on Machine Learning.

**Questions:**

**1. Clarification on the Gap with C-Adapter**

The core idea of your paper, learning a post-hoc, order-preserving transformation on classifier outputs to improve conformal prediction efficiency, appears to have a significant conceptual overlap with the recently proposed C-Adapter (Liu et al., 2024). This prior work is not discussed, making it difficult to assess the novelty of your contribution.

*   Could you please articulate the key conceptual and technical differences between FPS and C-Adapter?
*   Your primary technical novelty seems to be the use of an exponentiated trigonometric polynomial to guarantee monotonicity. What are the specific advantages of this parameterization over the constrained MLP often used in C-Adapter? Are there theoretical benefits (e.g., better approximation power, tighter generalization bounds) or practical advantages (e.g., optimization stability, reduced hyperparameter sensitivity) that justify this new approach?
*   To substantiate these claims, a direct empirical comparison, even on a single benchmark, would be highly valuable. Could you provide such a comparison or a compelling argument for why FPS is a significant advance over C-Adapter?

**2. On the Choice of the Order-Preserving Function Class**

The choice to parameterize the transformation via an exponentiated trigonometric polynomial is an interesting design choice that provides a structural guarantee of monotonicity. However, this specific parameterized class $\mathcal{G}$ is a strict subset of the space of all continuous, differentiable, and monotone functions $\mathcal{F}$.

*   Could you elaborate on the trade-offs made with this choice? Specifically, what is the potential "approximation error" (as implied by $\delta_M$ in Theorem 2) introduced by restricting the search space to $\mathcal{G}$?
*   Is it possible that a different, perhaps more flexible, function class, like monotone splines or a constrained neural network, could achieve a better trade-off between approximation power (capturing a more optimal transformation) and estimation variance (overfitting to the tuning set)?
*   And, what specific optimization challenges does your parameterization solve that might arise with other methods for enforcing monotonicity (e.g., projection, weight clipping, or complex reparameterizations)? A deeper discussion on why this specific class was chosen over others would strengthen the paper.

---

> ### Author Response · Authors · 2025-11-24
>
> We sincerely thank the reviewer for the time and for providing such detailed, constructive feedback. We are especially grateful for the pointers to the highly relevant work on C-Adapter and the literature on order-preserving functions. We offer the following responses to your specific questions.
>
> ### 1. Clarification on the Gap with C-Adapter
>
> We acknowledge the shared goal with C-Adapter (Liu et al., 2024): to learn a post-hoc transformation that improves the efficiency of conformal prediction, ideally while preserving the model's top-k accuracy. However, we wish to highlight two fundamental differences that motivate our work FPS and establish its novelty:
>
> - **Function Space and Expressiveness:** The two methods explore different function spaces: C-Adapter operates in a more restricted space and, for example, cannot represent the **identity function** ($f(x)=x$), which corresponds to the case where the original classifier is nearly optimal. In contrast, our FPS parameterization with an exponentiated trigonometric polynomial can learn this identity mapping ($\mathrm{a}=\mathrm{0}$), indicating a richer function space and greater flexibility across models and datasets.
> - **Theoretical Tractability and Interpretability:** Our goal is a method that is both empirically effective and **theoretically tractable**. C-Adapter relies on an MLP, whose generalization behavior is hard to analyze rigorously, whereas our interpretable parameterization is amenable to precise analysis and directly enables our main theoretical results, the **generalization bound (Theorem 2)**, providing a deeper understanding and stronger guarantees than existing MLP-based approaches.
>
>
>
> ### 2. On the Choice of the Order-Preserving Function Class
>
> We agree that there is an approximation–estimation trade-off. As shown in Theorem 2, the excess risk bound contains an estimation term of order $e^{A} D / \sqrt{n}$ and an approximation term $L_K \delta_M$. The quantity $\delta_M$ decreases with the number of Fourier modes $M$, so richer classes reduce approximation error. However, if we constrain the parameters via an $\ell_2$ bound $||a||_2 \le A$, then for $M$ coefficients a natural scaling gives $A = O(\sqrt{M})$, so $e^{A}$ (and thus the estimation term) grows with $M$. Enlarging the class therefore lowers $\delta_M$ but increases the estimation error, and Theorem 2 makes this trade-off explicit.
>
> We choose the exponentiated trigonometric polynomial as a low-complexity yet expressive class that empirically balances this trade-off while structurally guaranteeing smoothness and monotonicity. Compared with monotone splines or constrained neural networks, our parameterization avoids complex projections or constraints, keeps optimization simple (unconstrained gradient descent), and still performs strongly in the limited tuning data regime considered in the paper.
>
> Once again, we thank the reviewer for this insightful and actionable feedback. We are confident that addressing these points will significantly strengthen our paper.

---

### Official Review · Reviewer_TJxr · 2025-10-29

**Soundness:** 1
**Presentation:** 2
**Contribution:** 2
**Rating:** 2
**Confidence:** 3

**Summary:**

The paper aims at decreasing the set size in conformal prediction through optimizing an order preserving transformation. They optimize the parameter of the function over a tuning set and then apply it to the main conformal prediction pipeline ensuring that the guarantee holds.

**Strengths:**

The problem of decreasing the set size is an interesting problem where many papers have contributed to. The authors try to solve it by optimization which I believe is a promising direction towards the problem.

**Weaknesses:**

**Unnecessarily complicated notations.** I believe the paper at some points are presented in a more complicated way than they really are. For instance I would directly write Eq. 1 with the quantile definition as there is no need for writing it following the notation of the conformal risk control setup.

**Major issue with the goal of the paper and evalutions.** The authors define an order-preserving transformation meaning that within the score of one input x, the rank of the scores are preserved — which by definition keeps the top label at the same rank and preserves the accuracy. Is this rank preserved between any pairs of x, and y — I mean does it hold between class 4 of one input and class 2 of another input, or is it only over classes for a single input? I guess this transformation preserves the rank only between the labels of one input, otherwise it should not effect the set size as all the comparisons remain the same.

Now assuming that the order between the classes of different inputs change, then the question is if we are ignoring the main motivation behind adaptive prediction sets (APS) to have an adaptive prediction set. Which informally means that the conditional coverage remains high. If that is not the case then one can argue if the threshold prediction sets (LAC) also results in a smaller prediction sets.

The only place I see any notion close to conditional coverage is size-stratified coverage violation, which is not a proper estimator for conditional coverage. Originally Romano et al 2020 use worst slice coverage which however far from accurate is a better indicator of whether there is a region in the dataset that gets systematically miscovered. Importantly even on SSCV metric (which is not looking at the conditional coverage properly)  the authors only compare APS + theirs to LAC. When you are applying a transformation to a score function (in this case APS and RAPS) the first thing to check is whether you are sacrificing adaptivity for a better set size, and for that I would expect the authors to compare APS and APS + theirs in table 9.

**Questions:**

1. Please provide comparison on worst slice (slab) coverage (see Romano et al 2020, appendix) between your method and APS.
2. Please compare your method and APS on the set size coverage violation metric.

---

> ### Author Response · Authors · 2025-11-24
>
> We sincerely thank the reviewer for the constructive feedback. Here is a summary of our clarifications:
>
> The key clarification is that our transformation is order-preserving only among the classes for a single input $x$, and it does not preserve the order across different inputs.
>
> Following the reviewer’s suggestion, we directly compared original APS/RAPS and APS/RAPS augmented with our FPS framework. Table A and B show that across all models and both $\alpha$ levels, APS+ours and RAPS+ours consistently reduce set size compared with APS and RAPS while keeping WSC and SSCV at similar levels in most cases, whereas LAC achieves the smallest sets but suffers from clearly worse SSCV and slightly lower WSC, indicating that our method does indeed reduce set size without significantly harming conditional coverage.
>
> **Table A:** Size, WSC, and SSCV on ImageNet for different conformal methods and their FPS variants with $\alpha = 0.05$ across models.
>
> | Model       | Method    | Size   | WSC   | SSCV  |
> | ----------- | --------- | ------ | ----- | ----- |
> | ResNeXt101  | APS       | 20.831 | 0.929 | 0.040 |
> | ResNeXt101  | APS+ours  | 11.066 | 0.926 | 0.033 |
> | ResNeXt101  | RAPS      | 9.826  | 0.928 | 0.029 |
> | ResNeXt101  | RAPS+ours | 7.398  | 0.926 | 0.037 |
> | ResNeXt101  | LAC       | 3.794  | 0.922 | 0.082 |
> | ResNet152   | APS       | 14.218 | 0.933 | 0.030 |
> | ResNet152   | APS+ours  | 8.735  | 0.929 | 0.024 |
> | ResNet152   | RAPS      | 11.211 | 0.932 | 0.032 |
> | ResNet152   | RAPS+ours | 7.661  | 0.929 | 0.028 |
> | ResNet152   | LAC       | 3.570  | 0.920 | 0.125 |
> | ResNet101   | APS       | 15.909 | 0.934 | 0.033 |
> | ResNet101   | APS+ours  | 9.988  | 0.931 | 0.030 |
> | ResNet101   | RAPS      | 11.891 | 0.934 | 0.035 |
> | ResNet101   | RAPS+ours | 8.419  | 0.929 | 0.041 |
> | ResNet101   | LAC       | 3.963  | 0.921 | 0.098 |
> | DenseNet161 | APS       | 17.004 | 0.930 | 0.031 |
> | DenseNet161 | APS+ours  | 10.497 | 0.926 | 0.025 |
> | DenseNet161 | RAPS      | 12.695 | 0.930 | 0.030 |
> | DenseNet161 | RAPS+ours | 9.567  | 0.926 | 0.030 |
> | DenseNet161 | LAC       | 4.213  | 0.920 | 0.120 |
> | VGG16       | APS       | 24.518 | 0.935 | 0.027 |
> | VGG16       | APS+ours  | 17.0   | 0.936 | 0.021 |
> | VGG16       | RAPS      | 16.623 | 0.937 | 0.027 |
> | VGG16       | RAPS+ours | 13.159 | 0.936 | 0.044 |
> | VGG16       | LAC       | 7.160  | 0.927 | 0.067 |
> | ShuffleNet  | APS       | 54.987 | 0.935 | 0.027 |
> | ShuffleNet  | APS+ours  | 31.794 | 0.934 | 0.018 |
> | ShuffleNet  | RAPS      | 37.935 | 0.934 | 0.035 |
> | ShuffleNet  | RAPS+ours | 26.024 | 0.933 | 0.043 |
> | ShuffleNet  | LAC       | 12.838 | 0.927 | 0.133 |
>
>
>
> **Table B:** Size, WSC, and SSCV on ImageNet for different conformal methods and their FPS variants with $\alpha = 0.10$ across models.
>
> | Model       | Method    | Size   | WSC   | SSCV  |
> | ----------- | --------- | ------ | ----- | ----- |
> | ResNeXt101  | APS       | 7.096  | 0.866 | 0.071 |
> | ResNeXt101  | APS+ours  | 4.824  | 0.863 | 0.054 |
> | ResNeXt101  | RAPS      | 4.753  | 0.864 | 0.075 |
> | ResNeXt101  | RAPS+ours | 3.746  | 0.860 | 0.080 |
> | ResNeXt101  | LAC       | 1.684  | 0.847 | 0.295 |
> | ResNet152   | APS       | 6.343  | 0.873 | 0.062 |
> | ResNet152   | APS+ours  | 4.608  | 0.870 | 0.046 |
> | ResNet152   | RAPS      | 4.561  | 0.873 | 0.076 |
> | ResNet152   | RAPS+ours | 3.727  | 0.870 | 0.075 |
> | ResNet152   | LAC       | 1.747  | 0.851 | 0.273 |
> | ResNet101   | APS       | 6.955  | 0.875 | 0.057 |
> | ResNet101   | APS+ours  | 4.979  | 0.873 | 0.053 |
> | ResNet101   | RAPS      | 4.752  | 0.875 | 0.081 |
> | ResNet101   | RAPS+ours | 3.838  | 0.873 | 0.071 |
> | ResNet101   | LAC       | 1.762  | 0.855 | 0.265 |
> | DenseNet161 | APS       | 6.910  | 0.872 | 0.067 |
> | DenseNet161 | APS+ours  | 5.134  | 0.870 | 0.055 |
> | DenseNet161 | RAPS      | 4.433  | 0.868 | 0.089 |
> | DenseNet161 | RAPS+ours | 3.835  | 0.867 | 0.079 |
> | DenseNet161 | LAC       | 1.917  | 0.847 | 0.329 |
> | VGG16       | APS       | 11.844 | 0.878 | 0.049 |
> | VGG16       | APS+ours  | 9.513  | 0.879 | 0.039 |
> | VGG16       | RAPS      | 10.215 | 0.880 | 0.035 |
> | VGG16       | RAPS+ours | 8.454  | 0.879 | 0.033 |
> | VGG16       | LAC       | 2.989  | 0.854 | 0.209 |
> | ShuffleNet  | APS       | 22.625 | 0.878 | 0.046 |
> | ShuffleNet  | APS+ours  | 16.116 | 0.878 | 0.032 |
> | ShuffleNet  | RAPS      | 15.554 | 0.878 | 0.025 |
> | ShuffleNet  | RAPS+ours | 12.225 | 0.876 | 0.027 |
> | ShuffleNet  | LAC       | 4.305  | 0.859 | 0.163 |
>
> Your feedback has helped us clarify our contribution and strengthen our evaluation.

---

### Official Review · Reviewer_r1Mf · 2025-10-30

**Soundness:** 3
**Presentation:** 2
**Contribution:** 2
**Rating:** 4
**Confidence:** 4

**Summary:**

This paper introduces Flexible Prediction Sets (FPS), a post-hoc framework that learns order-preserving transformations of predicted class probabilities to reduce conformal prediction set sizes while maintaining coverage guarantees.

**Strengths:**

Comprehensive theoretical analysis including coverage preservation (Theorem 1), generalization bounds with approximation and estimation error decomposition.

Experiments across ImageNet, ImageNet-V2, and Banking77 with multiple base architectures demonstrate consistent set size reductions.

**Weaknesses:**

The fundamental insight of transforming scores before conformal prediction overlaps significantly with existing calibration methods like temperature scaling. Using sigmoid function to construct a differentiable set size loss already exists in conformal training literatures. While order-preservation is a useful constraint, the core contribution is incremental.

Requires separate tuning dataset and hyperparameter optimization $M, \gamma, \beta$, adding complexity compared to standard CP methods. The need for a held-out tuning set may be prohibitive in data-scarce scenarios.

The paper describes FPS as a "post-hoc and computationally light" framework but provides no explicit quantitative measures of the time required to learn the transformation $g_a$.

The exponential trigonometric polynomial parameterization appears tedious given that established monotonic models already exist in the literature. The authors justify this choice for "structural guarantee," "approximation power," and "optimization stability" but don't adequately compare against simpler alternatives like monotonic splines or constrained neural networks.

The transformation framework appears specifically tailored to APS-type cumulative scoring functions, raising questions about its applicability to other conformal prediction methods. While the authors briefly mention extension to RAPS, the method's compatibility with other important scoring functions (e.g., margin-based scores, entropy-based scores, or model-specific scores) remains unexplored.

Assumption 2 (vanishing loss update) appears restrictive with limited empirical justification beyond Appendix C. The assumption that calibration component updates remain small may not hold generally across different base models and datasets.

Focus primarily on marginal coverage with limited conditional coverage analysis. The brief SSCV analysis in Appendix F shows some degradation in group conditional coverage, which could be concerning for fairness-sensitive applications.

**Questions:**

The authors should address the concerns raised in the weakness section above.

---

> ### Author Response · Authors · 2025-11-24
>
> Thank you for your valuable feedback and insightful comments.  We would like to offer the following clarifications and responses:
>
> ### 1. Novelty and Overlap with Existing Methods
>
> We acknowledge that our work shares a common goal with existing methods of improving efficiency of predictions. However, we would like to emphasize that a shared goal does not imply a lack of novelty in the method itself. Our core contribution lies in proposing this optimizable transformation framework. As Table A demonstrates, compared to Temperature Scaling, our method shows a more significant advantage in effectively reducing the prediction set size.
>
> **Table A:** Size results on ImageNet across $\alpha$ levels and base image classifiers. APS+TS / RAPS+TS apply temperature scaling; APS and RAPS use a fixed temperature of 1; APS+ours / RAPS+ours apply our FPS framework.
>
> | Model       | $\alpha$ | APS+TS | APS    | APS+ours | RAPS+TS | RAPS   | RAPS+ours |
> | ----------- | -------- | ------ | ------ | -------- | ------- | ------ | --------- |
> | ResNeXt101  | 0.05     | 45.083 | 20.865 | 10.939   | 4.316   | 3.829  | 3.640     |
> | ResNeXt101  | 0.10     | 19.301 | 7.171  | 2.894    | 2.603   | 2.020  | 1.966     |
> | ResNet152   | 0.05     | 22.958 | 14.725 | 8.298    | 4.559   | 4.087  | 4.032     |
> | ResNet152   | 0.10     | 10.462 | 6.360  | 3.010    | 2.618   | 2.260  | 2.176     |
> | ResNet101   | 0.05     | 24.126 | 16.091 | 9.022    | 4.795   | 4.417  | 4.382     |
> | ResNet101   | 0.10     | 10.917 | 7.015  | 3.315    | 2.726   | 2.387  | 2.286     |
> | DenseNet161 | 0.05     | 28.308 | 17.218 | 9.866    | 5.164   | 4.702  | 4.664     |
> | DenseNet161 | 0.10     | 12.343 | 6.956  | 3.275    | 2.775   | 2.338  | 2.299     |
> | VGG16       | 0.05     | 27.327 | 23.917 | 15.329   | 8.268   | 8.803  | 8.542     |
> | VGG16       | 0.10     | 13.705 | 11.845 | 5.943    | 4.086   | 3.768  | 3.577     |
> | ShuffleNet  | 0.05     | 70.568 | 54.133 | 27.588   | 15.319  | 15.696 | 15.029    |
> | ShuffleNet  | 0.10     | 32.190 | 22.584 | 8.931    | 5.626   | 5.026  | 4.898     |
>
>
>
> ### 2. Tuning Complexity, Additional Data, and Computational Overhead
>
> In data-scarce settings, we agree that holding out an extra tuning set can be problematic, so in Appendix D we additionally study strategies that avoid this, such as optimizing directly on the calibration set, and find that our method remains effective without a separate tuning set. Moreover, the optimization itself is computationally lightweight: on an Intel Xeon CPU (12 cores) and an NVIDIA GeForce GTX 1080 Ti GPU, one training pass of the transformation (FPS training) on ImageNet takes only about 30 seconds, so even a simple grid search keeps the overall cost modest and does not pose a practical barrier to implementation.
>
> ### 3. Choice of Parametrization
>
> We use the exponential trigonometric polynomial because, although alternatives such as splines or constrained monotonic neural networks are viable, they typically require more parameters for comparable flexibility and enforcing their monotonicity turns the problem into a constrained optimization task, which is considerably more difficult and computationally demanding. In contrast, our parametrization guarantees monotonicity by construction, so we can use unconstrained optimization, leading to the optimization stability and ease of solving that we aimed for.
>
> ### 4. Applicability to Other Scoring Functions
>
> Our current work focuses on APS and RAPS, which are among the most widely used scoring functions. The core idea of FPS is to learn an order-preserving transformation that optimizes the score distribution for set size reduction, and our APS/RAPS results suggest that this approach can be extended to other scoring functions in future work.
>
> ### 5. Restrictiveness of Assumption 2
>
> We introduced Assumption 2 mainly for theoretical rigor, and while it may not strictly hold in all scenarios, the experiments in Appendix C show that updates to the calibration component are very small in our settings, so the assumption is practically satisfied and does not undermine the empirical strength of our results.
>
> ### 6. Conditional Coverage
>
> Our primary goal in this paper was to improve prediction set efficiency under strict marginal coverage, while our experiments show that conditional coverage is only mildly affected, as reflected by WSC and SSCV metrics.
>
> Once again, we sincerely thank you for your valuable feedback. We hope these clarifications help illustrate the contributions and context of our work.

---

> > ### Author Response · Authors · 2025-11-24
> >
> > Additionally, Table B and C show that across all models and both $\alpha$ levels, APS+ours and RAPS+ours consistently reduce set size compared with APS and RAPS while keeping WSC and SSCV at similar levels in most cases, whereas LAC achieves the smallest sets but suffers from clearly worse SSCV and slightly lower WSC, indicating a stronger loss of conditional coverage.
> >
> > **Table B:** Size, WSC, and SSCV on ImageNet for different conformal methods and their FPS variants with $\alpha = 0.05$ across models.
> >
> > | Model       | Method    | Size   | WSC   | SSCV  |
> > | ----------- | --------- | ------ | ----- | ----- |
> > | ResNeXt101  | APS       | 20.831 | 0.929 | 0.040 |
> > | ResNeXt101  | APS+ours  | 11.066 | 0.926 | 0.033 |
> > | ResNeXt101  | RAPS      | 9.826  | 0.928 | 0.029 |
> > | ResNeXt101  | RAPS+ours | 7.398  | 0.926 | 0.037 |
> > | ResNeXt101  | LAC       | 3.794  | 0.922 | 0.082 |
> > | ResNet152   | APS       | 14.218 | 0.933 | 0.030 |
> > | ResNet152   | APS+ours  | 8.735  | 0.929 | 0.024 |
> > | ResNet152   | RAPS      | 11.211 | 0.932 | 0.032 |
> > | ResNet152   | RAPS+ours | 7.661  | 0.929 | 0.028 |
> > | ResNet152   | LAC       | 3.570  | 0.920 | 0.125 |
> > | ResNet101   | APS       | 15.909 | 0.934 | 0.033 |
> > | ResNet101   | APS+ours  | 9.988  | 0.931 | 0.030 |
> > | ResNet101   | RAPS      | 11.891 | 0.934 | 0.035 |
> > | ResNet101   | RAPS+ours | 8.419  | 0.929 | 0.041 |
> > | ResNet101   | LAC       | 3.963  | 0.921 | 0.098 |
> > | DenseNet161 | APS       | 17.004 | 0.930 | 0.031 |
> > | DenseNet161 | APS+ours  | 10.497 | 0.926 | 0.025 |
> > | DenseNet161 | RAPS      | 12.695 | 0.930 | 0.030 |
> > | DenseNet161 | RAPS+ours | 9.567  | 0.926 | 0.030 |
> > | DenseNet161 | LAC       | 4.213  | 0.920 | 0.120 |
> > | VGG16       | APS       | 24.518 | 0.935 | 0.027 |
> > | VGG16       | APS+ours  | 17.0   | 0.936 | 0.021 |
> > | VGG16       | RAPS      | 16.623 | 0.937 | 0.027 |
> > | VGG16       | RAPS+ours | 13.159 | 0.936 | 0.044 |
> > | VGG16       | LAC       | 7.160  | 0.927 | 0.067 |
> > | ShuffleNet  | APS       | 54.987 | 0.935 | 0.027 |
> > | ShuffleNet  | APS+ours  | 31.794 | 0.934 | 0.018 |
> > | ShuffleNet  | RAPS      | 37.935 | 0.934 | 0.035 |
> > | ShuffleNet  | RAPS+ours | 26.024 | 0.933 | 0.043 |
> > | ShuffleNet  | LAC       | 12.838 | 0.927 | 0.133 |
> >
> >
> >
> > **Table C:** Size, WSC, and SSCV on ImageNet for different conformal methods and their FPS variants with $\alpha = 0.10$ across models.
> >
> > | Model       | Method    | Size   | WSC   | SSCV  |
> > | ----------- | --------- | ------ | ----- | ----- |
> > | ResNeXt101  | APS       | 7.096  | 0.866 | 0.071 |
> > | ResNeXt101  | APS+ours  | 4.824  | 0.863 | 0.054 |
> > | ResNeXt101  | RAPS      | 4.753  | 0.864 | 0.075 |
> > | ResNeXt101  | RAPS+ours | 3.746  | 0.860 | 0.080 |
> > | ResNeXt101  | LAC       | 1.684  | 0.847 | 0.295 |
> > | ResNet152   | APS       | 6.343  | 0.873 | 0.062 |
> > | ResNet152   | APS+ours  | 4.608  | 0.870 | 0.046 |
> > | ResNet152   | RAPS      | 4.561  | 0.873 | 0.076 |
> > | ResNet152   | RAPS+ours | 3.727  | 0.870 | 0.075 |
> > | ResNet152   | LAC       | 1.747  | 0.851 | 0.273 |
> > | ResNet101   | APS       | 6.955  | 0.875 | 0.057 |
> > | ResNet101   | APS+ours  | 4.979  | 0.873 | 0.053 |
> > | ResNet101   | RAPS      | 4.752  | 0.875 | 0.081 |
> > | ResNet101   | RAPS+ours | 3.838  | 0.873 | 0.071 |
> > | ResNet101   | LAC       | 1.762  | 0.855 | 0.265 |
> > | DenseNet161 | APS       | 6.910  | 0.872 | 0.067 |
> > | DenseNet161 | APS+ours  | 5.134  | 0.870 | 0.055 |
> > | DenseNet161 | RAPS      | 4.433  | 0.868 | 0.089 |
> > | DenseNet161 | RAPS+ours | 3.835  | 0.867 | 0.079 |
> > | DenseNet161 | LAC       | 1.917  | 0.847 | 0.329 |
> > | VGG16       | APS       | 11.844 | 0.878 | 0.049 |
> > | VGG16       | APS+ours  | 9.513  | 0.879 | 0.039 |
> > | VGG16       | RAPS      | 10.215 | 0.880 | 0.035 |
> > | VGG16       | RAPS+ours | 8.454  | 0.879 | 0.033 |
> > | VGG16       | LAC       | 2.989  | 0.854 | 0.209 |
> > | ShuffleNet  | APS       | 22.625 | 0.878 | 0.046 |
> > | ShuffleNet  | APS+ours  | 16.116 | 0.878 | 0.032 |
> > | ShuffleNet  | RAPS      | 15.554 | 0.878 | 0.025 |
> > | ShuffleNet  | RAPS+ours | 12.225 | 0.876 | 0.027 |
> > | ShuffleNet  | LAC       | 4.305  | 0.859 | 0.163 |

---

### Note · Authors · 2025-11-24

I have read and agree with the venue's withdrawal policy on behalf of myself and my co-authors.